# Antagonistic nanobodies implicate mechanism of GSDMD pore formation and potential therapeutic application

Lisa D. J. Schiffelers [1], Yonas M. Tesfamariam [1], Lea-Marie Jenster[1], Stefan Diehl [1], Sophie C. Binder [1], Sabine Normann[1], Jonathan Mayr [1], Steffen Pritzl[1], Elena Hagelauer [1], Anja Kopp [2,3], Assaf Alon [4], Matthias Geyer [2], Hidde L. Ploegh [4] & Florian I. Schmidt [1,4,5] ✉

Inflammasome activation results in the cleavage of gasdermin D (GSDMD) by pro-inflammatory caspases. The N-terminal domains (GSDMD[NT]) oligomerize and assemble pores penetrating the target membrane. As methods to study pore formation in living cells are insufficient, the order of conformational changes, oligomerization, and membrane insertion remained unclear. We have raised nanobodies (VHHs) against human GSDMD and find that cytosolic expression of VHH[GSDMD-1] and VHH[GSDMD-2] prevents oligomerization of GSDMD[NT] and pyroptosis. The nanobody-stabilized GSDMD[NT] monomers partition into the plasma membrane, suggesting that membrane insertion precedes oligomerization. Inhibition of GSDMD pore formation switches cell death from pyroptosis to apoptosis, likely driven by the enhanced caspase-1 activity required to activate caspase-3. Recombinant antagonistic nanobodies added to the extracellular space prevent pyroptosis and exhibit unexpected therapeutic potential. They may thus be suitable to treat the ever-growing list of diseases caused by activation of (non-) canonical inflammasomes.

Gasdermin D (GSDMD) is a pore forming protein that perforates the plasma membrane to execute pyroptotic cell death. It is thus considered the key effector protein of the inflammasome pathway[1–4]. Canonical inflammasomes are multiprotein complexes comprised of sensor proteins that oligomerize and recruit the adapter protein ASC as well as pro-inflammatory caspase-1[5]. Distinct inflammasome sensors are activated by different pathogen- or danger-associated molecular patterns (DAMPs). Human NAIP/NLRC4 is activated when NAIP binds to the needle proteins of bacterial type III secretion systems, such as *Shigella flexneri MxiH*, and subsequently initiates oligomerization of NLRC4[6]. NLRP3 is an indirect sensor for potassium efflux and perturbations of intracellular homeostasis[7]. The ensuing activation of caspase-1 is not only responsible for the maturation of pro-inflammatory cytokines IL-1β and IL-18, but also for the cleavage of

GSDMD in its interdomain linker. This releases the N-terminus (GSDMD[NT]) from the control of the autoinhibitory C-terminus (GSDMD[CT]), allowing GSDMD[NT] to assembles pores in the plasma membrane[1–3]. As a result, the plasma membrane becomes permeable to DNA intercalating dyes such as propidium iodide or DRAQ7, and the mature cytokines IL-1β and IL-18 are released[1,3,8]. Eventually, the entire cell ruptures and releases larger cytosolic components into the cellular environment, including tetrameric lactate dehydrogenase (LDH) and pro-inflammatory DAMPs that further promote inflammation. Membrane rupture itself seems to depend on the cell-surface protein Ninjurin-1, which oligomerizes in the plasma membrane after GSDMD pore formation[9].

The structures of soluble full-length GSDMD and GSDMD[NT] pores were elucidated by X-ray crystallography and electron microscopy[10,11].

[1]Institute of Innate Immunity, Medical Faculty, University of Bonn, Bonn, Germany. [2]Institute of Structural Biology, Medical Faculty, University of Bonn, Bonn, Germany. [3]Inflammation Division, The Walter and Eliza Hall Institute of Medical Research, Parkville, Australia. [4]Whitehead Institute for Biomedical Research, Cambridge, MA, USA. [5]Core Facility Nanobodies, Medical Faculty, University of Bonn, Bonn, Germany. ✉e-mail: fschmidt@uni-bonn.de

GSDMD pores reconstituted in vitro are composed of 31-34 monomers, forming a pore with an estimated inner diameter of 22 nm[11]. After cleavage, GSDMD[NT] undergoes drastic conformational changes: two extension domains, composed of short beta sheets and helices, transform into two beta hairpins with extended beta sheets, which constitute the membrane-spanning pore upon oligomerization[10,12]. It remained unclear if loss of GSDMD[CT] is sufficient to trigger these conformational changes, or whether they only occur in concert with oligomerization. Atomic force microscopy suggests that GSDMD[NT] forms oligomers of different sizes in artificial membranes; smaller slits or arcs were observed to grow into symmetric pores[13]. On the other hand, pore-like structures of human GSDMD and murine GSDMA3 composed of oligomerized globular GSDMD[NT]/GSDMA3[NT] protomers that do not penetrate the target membrane were observed in samples of purified pores by electron microscopy and on supported lipid membranes by atomic force microscopy[11,12,14]. The authors thus speculated that oligomerization of a prepore precedes coordinated conformational changes of all subunits that lead to membrane penetration[11,12]. Interestingly, GSDMD pore formation may also be further regulated: Efficient GSDMD pore formation relies on ROS generated by the Ragulator-Rag-mTORC1 pathway, which likely mediates the oxidative post-translational modification of GSDMD cysteine 191[15,16]. The same residue has recently been proposed to undergo (reversible) palmitoylation to facilitate membrane association and full activation[17-19].

Apart from assays reporting the permeability of the plasma membrane to different dyes or cell death, pore formation of endogenous GSDMD had not been observed in molecular detail in living cells, largely due to the lack of suitable tools[8]. Pyroptotic cells are very delicate and are not compatible with staining methods involving fixation and multiple washing steps. Moreover, upon inflammasome activation, fluorescent derivatives of GSDMD[NT] were barely observed in the plasma membrane, but mostly in intracellular compartments or structures[20].

To provide more insights into GSDMD pore formation in live cells, we generated nanobodies against the human GSDMD protein. Nanobodies are single domain antibodies derived from the variable domain of heavy chain-only antibodies (VHH) present in *camelids*[21]. Due to their small size, specificity, and functionality in the cytosol, they present themselves as useful tools to study target proteins in living cells[22]. We identified two antagonistic GSDMD nanobodies that inhibit pyroptosis and IL-1β release by blocking oligomerization of GSDMD[NT]. As nanobody-bound GSDMD[NT] still partitions into the plasma membrane, we conclude that monomeric GSDMD[NT] exhibits a suitable conformation to insert into the plasma membrane and only oligomerizes after insertion. We describe an additional layer of negative caspase-1 regulation by functional GSDMD pores and find that the inhibitory nanobodies show great potential in preventing inflammatory cell death when administered to the extracellular environment. This is of particular interest since GSDMD is linked to an ever-growing list of (auto)inflammatory, metabolic, and neurodegenerative diseases and cancer, and is thus an eminent drug target[23-25].

## Results

### Identification of GSDMD-specific nanobodies
To study GSDMD pore formation in living cells, we raised nanobodies against the human GSDMD protein[21]. An alpaca (*Vicugna pacos*) was immunized with recombinant full-length GSDMD and six hits were identified by phage display (Figs. 1A, B, Figure S1A). ELISA experiments confirmed their specificity for GSDMD (Fig. 1C). LUMIER assays confirmed binding of VHH[GSDMD-1], VHH[GSDMD-2], VHH[GSDMD-3] and VHH[GSDMD-5] to GSDMD in the cytosol (Fig. 1D), of which VHH[GSDMD-1] and VHH[GSDMD-2] clearly recognize the N-terminal domain of GSDMD. None of the nanobodies bound to murine GSDMD in the stringent conditions of the LUMIER assay (Figure S1B).

### VHH[GSDMD-1] and VHH[GSDMD-2] abrogate pyroptosis
We investigated whether the identified nanobodies perturb GSDMD function if expressed intracellularly. Interestingly, VHH[GSDMD-1] and to some extent VHH[GSDMD-2] inhibited the release of LDH in HEK293T cells overexpressing GSDMD[NT], but not GSDME[NT] (Figs. 2A, B). We next generated human myeloid THP-1 cell lines constitutively expressing the HA-tagged nanobodies. THP-1 WT cells and cells expressing an unrelated nanobody against the nucleoprotein of influenza A virus (VHH[NP-1])[26] were used as negative controls. Previously identified VHH[ASC] interferes with inflammasome formation and IL-1β release and served as positive control[27,28]. VHH[GSDMD-1] and VHH[GSDMD-2] were expressed at levels similar to the control nanobodies, while VHH[GSDMD-3] was poorly expressed and excluded from further analysis (Figure S2A). PMA-differentiated THP-1 cells were activated with the *Shigella* needle protein MxiH to induce NLRC4 inflammasome activation[6], or with LPS and nigericin to activate the NLRP3 inflammasome[29]. VHH[GSDMD-1] and VHH[GSDMD-2] completely shut down the release of LDH (Figs. 2C, D) and IL-1β (Figs. 2E, F), and likewise prevented the uptake of the membrane-impermeable DNA dye DRAQ7 and pyroptotic morphology (Fig. 2G, Figure S2B). Primary human macrophages were transduced with lentivirus encoding the nanobody of interest in addition to our previously described fluorescent inflammasome reporter caspase-1[CARD]-EGFP (C1C-EGFP)[30] (Figs. 2H, I, Figure S2C). Upon treatment with MxiH, we observed a strong reduction in cell counts, as pyroptotic cells are too fragile to survive processing for flow cytometry (Fig. 2J, Figure S2D). Yet, macrophages expressing antagonistic VHH[GSDMD-1] or VHH[GSDMD-2] and C1C-EGFP preferentially survived, as EGFP-positive cells were enriched after treatment (Fig. 2I, Figure S2C). Detection of C1C-EGFP-positve ASC specks by flow cytometry revealed that NLRC4 inflammasomes were robustly assembled, indicating that cells survived because pyroptosis downstream of inflammasomes was inhibited (Fig. 2K, Figure S2E).

### Antagonistic nanobodies prevent oligomerization of GSDMD[NT], but do not impair inflammasome assembly or GSDMD cleavage
To elucidate the mechanism of pyroptosis inhibition, we first ruled out any effect of GSDMD nanobodies on ASC speck assembly (Fig. 2L, Figure S2F). No differences in GSDMD expression and cleavage were detected by immunoblot (Fig. 2M). It had previously been shown that GSDMD oligomers appeared as high-molecular weight bands by SDS-PAGE under non-reducing conditions[31,32]. We thus analyzed GSDMD bands by immunoblot and found that said dimers and higher order oligomers disappeared in the lysates from VHH[GSDMD-1]- or VHH[GSDMD-2]-expressing cells, indicating that the nanobodies interfere with the oligomerization of GSDMD[NT] (Fig. 2N).

### Nanobodies preventing oligomerization still allow membrane localization of GSDMD[NT]
Having established a system in which antagonistic nanobodies stabilize monomeric GSDMD[NT] by preventing oligomerization, we were curious if the monomeric protein would be sufficient to insert into membranes. We therefore transfected HEK293T cells stably expressing VHH-EGFP fusions with expression vectors for full-length GSDMD-mCherry or GSDMD[NT]-mCherry and followed the localization by live cell confocal microscopy. Initial experiments were performed with GSDMD mutant I104N, which was reported to facilitate observation of GSDMD[NT] in the plasma membrane[2,4,33]. In the presence of VHH[NP-1], full-length GSDMD did not insert into the plasma membrane as expected (Fig. 3A), and GSDMD[NT]-mCherry was mostly found in internal structures of the resulting pyroptotic cells (Fig. 3A, yellow arrows). This suggests that pores of GSDMD[NT] do not accumulate in the plasma membrane and are possibly rapidly removed, e.g., by membrane repair processes[34]. When GSDMD[NT] was co-expressed in cells with VHH[GSDMD-1] or VHH[GSDMD-2], GSDMD[NT]-mCherry almost completely partitioned into the rim of the cell, where it co-localized with VHH-EGFP (Fig. 3A). The cells no longer showed signs of pyroptosis. Co-localization of WT

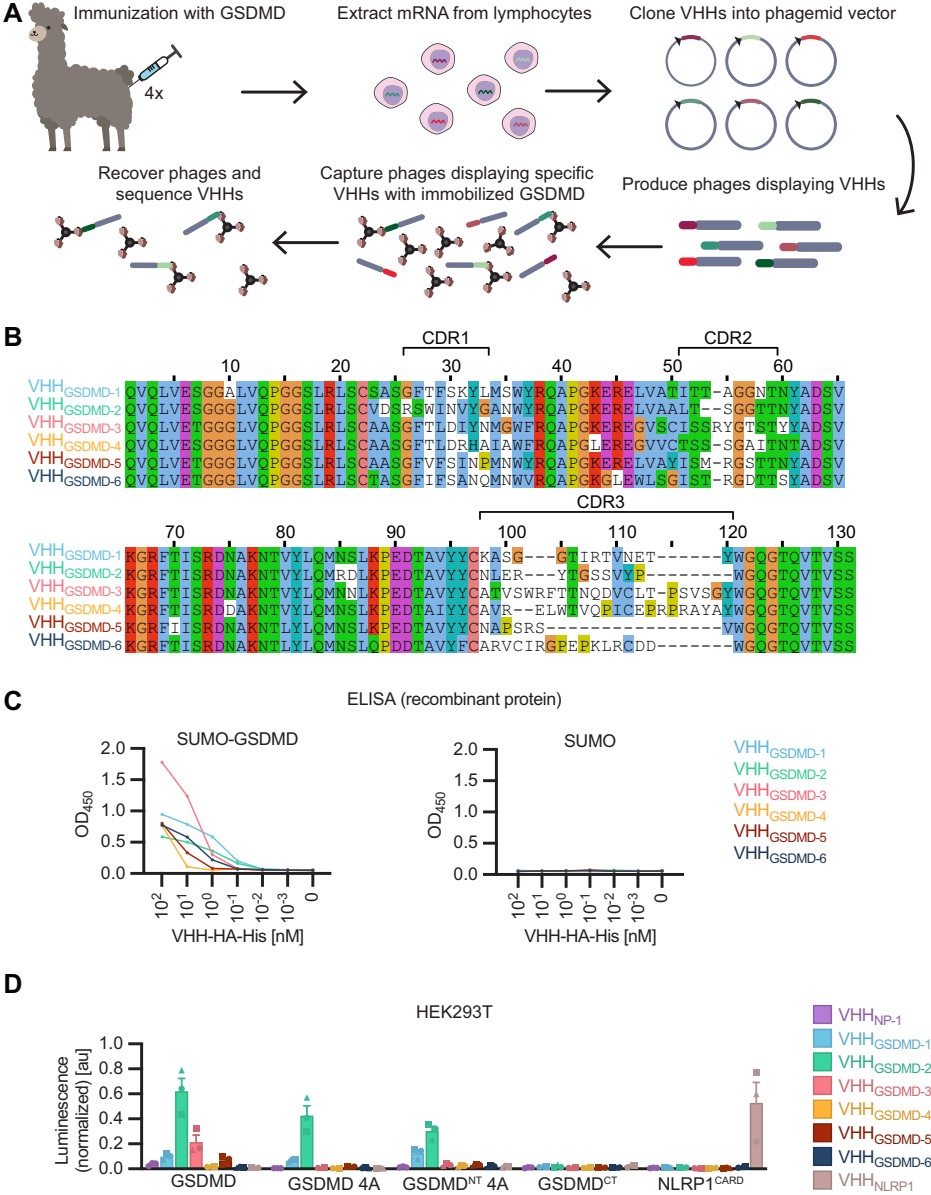

**Fig. 1 | Identification of GSDMD-specific nanobodies. A** Scheme of alpaca immunization and GSDMD nanobody (VHH) selection by phage display. **B** Sequence alignment of the 6 GSDMD-specific nanobodies, with indication of the complementarity determining regions (CDRs). **C** ELISA with recombinant nanobodies: SUMO-GSDMD or control protein SUMO was immobilized on ELISA plates and binding of the indicated concentration of HA-His-tagged nanobodies was quantified by ELISA with anti-HA HRP. **D** LUMIER assay: HEK293T cells were co-transfected with expression vectors for the specified HA-tagged nanobodies and the indicated protein-Renilla luciferase fusions. 24 h post transfection, cells were lysed, and VHH-HA was immunoprecipitated with immobilized anti-HA. Coelenterazine-h was added and luminescence of co-purified Renilla luciferase was measured and normalized to luminescence of lysates. Data represent average values (with individual data points) from three independent experiments ± SEM. The vector graphic of the alpaca shown in the schematic in Fig. 1A was designed by Freepik.

GSDMT$^{NT}$-mCherry with the plasma membrane marker emiRFP-CAAX confirmed localization to the plasma membrane and ruled out that any findings were only specific to the I104N mutation (Fig. 3B). Therefore, we hypothesize that monomeric nanobody-bound GSDMD$^{NT}$ inserts into the plasma membrane and that the required conformational changes for membrane integration occur in monomeric GSDMD after removal of the auto-inhibitory C-terminus. It is challenging to prove that GSDMD$^{NT}$-mCherry is genuinely inserted into the plasma membrane and not merely associated with the plasma membrane through the basic patches BP1, BP2, and BP3 defined by Xia et al.[13]. We therefore analyzed the mutants C191A and C191S which should only interfere with insertion, as C191 is found in the tip of the extended beta sheets of GSDMD$^{NT}$ that inserts into the membrane. GSDMD$^{NT}$-mCherry C191A and C191S were severely impaired in plasma membrane partitioning

and fluorescence was found either only in the cytosol or in both the cytosol and the plasma membrane (Fig. 3B, Figure S3A). We therefore conclude that GSDMD$^{NT}$-mCherry requires membrane insertion to achieve the complete redistribution into the plasma membrane that we observed.

To test if the antagonistic nanobodies also stabilized GSDMD$^{NT}$ after inflammasome activation, we generated THP-1 cells expressing HA-tagged VHHs in addition to GSDMD with mNeonGreen (mNG) inserted after amino acid 270 (GSDMD-mNG_ins), i.e., before the caspase-1 cleavage site (Figs. 4A, B)[20]. We did not observe GSDMD accumulation in the plasma membrane in pyroptotic THP-1 cells in presence of VHH$_{NP-1}$ either. Most of GSDMD$^{NT}$-mNG generated by caspase-1 cleavage localized to internal structures as observed for the HEK293T cells (Fig. 4A, top right, yellow arrows). When inflammasome

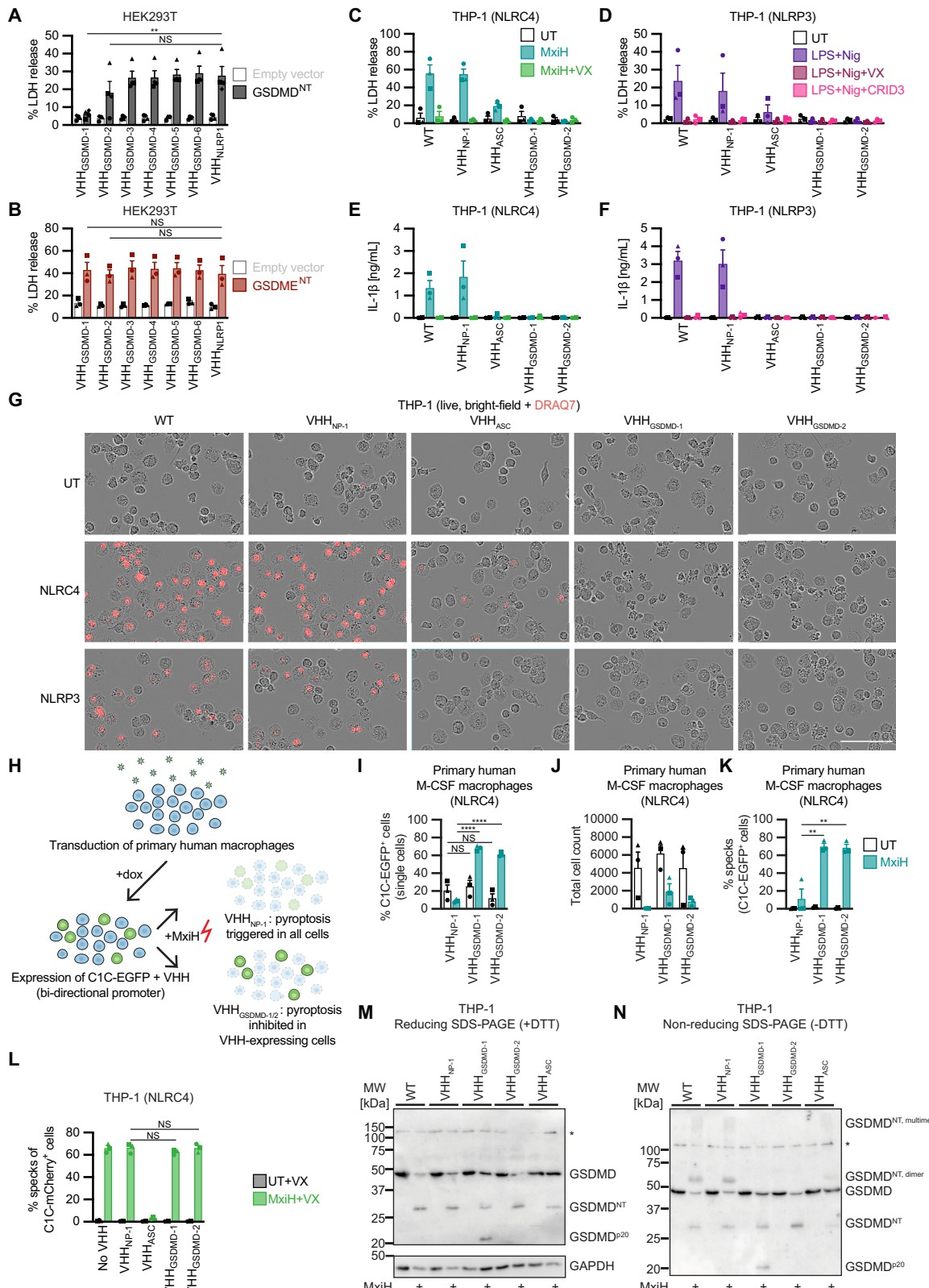

activation was triggered in presence of VHH_GSDMD-1 or VHH_GSDMD-2, cells did not exhibit morphological features of pyroptosis and GSDMD^NT-mNG co-localized with CellMask Orange (CMO) plasma membrane stain (Figs. 4A, C, D, Figure S3B). No membrane localization of GSDMD-mNG was observed for cells treated with the pore-forming toxin perfringolysin O (PFO) from *Clostridium perfringens*, indicating that the observed membrane association was not merely explained by

the loss of cytosolic GSDMD-mNG through pores and exposure of a pre-existing membrane-associated pool of GSDMD. We also followed GSDMD-mNG_ins in CMO-stained THP-1 cells by live cell confocal microscopy (Figure S4A, movies S1, and S2) and observed that the total mNG fluorescence dropped during pyroptotic swelling, perhaps due to the loss of soluble uncleaved GSDMD-mNG_ins. As before, very little green fluorescence was observed in the plasma membrane of

**Fig. 2 | VHH$_{GSDMD-1}$ and VHH$_{GSDMD-2}$ abrogate pyroptosis by interfering with GSDMD$^{NT}$ oligomerization. A, B** HEK293T cells were co-transfected with expression vectors for the indicated HA-tagged nanobodies as well as empty vector, GSDMD$^{NT}$ (**A**), or GSDME$^{NT}$ (**B**). LDH release was measured 24 h post transfection and normalized to cells lysed in 1% Triton X-100 ($n = 4$ [A] or $n = 3$ [B] biological replicates). **C–F** PMA-differentiated THP-1 macrophages constitutively expressing the indicated HA-tagged nanobodies or WT controls were stimulated with 1.0 µg/mL PA and 0.1 µg/mL LFn-MxiH (MxiH) for 1 h to activate NLRC4 (**C, E**), or with 200 ng/mL ultrapure LPS for 3 h and 10 µM nigericin (Nig) for 1 h to activate NLRP3 (**D, F**), in the presence of 40 µM VX-765 (VX) or 2.5 µM CRID3 where indicated. LDH release was measured as in **A** and **B** (**C, D**), and IL-1β in the supernatant was measured by Homogeneous Time Resolved Fluorescence (HTRF) (**E, F**). **G** PMA-differentiated THP-1 macrophages were stimulated with NLRC4 and NLRP3 activators as described above, but in the presence of 100 nM DRAQ7. DRAQ7 uptake was monitored over 5 h in an Incucyte Live-Cell Imaging system. Representative images (of $n = 3$) after 1 h of normalized DRAQ7 uptake are displayed. Scale bar, 100 µm. Quantified DRAQ7 uptake over time from the same experiment is displayed in Figure S2B. **H** Overview of transduction of primary human macrophages with lentivirus particles packaging Vpx-Vpx and encoding C1C-EGFP and the different nanobodies under the control of a bi-directional doxycycline (dox)-inducible promoter. Stimulation with NLRC4 activator MxiH triggers cell death by pyroptosis,

unless the expressed nanobodies inhibit GSDMD pore formation, which leads to the enrichment of the respective transduced (C1C-EGFP-positive) cells. **I–K** Primary M-CSF-differentiated monocyte-derived human macrophages were transduced with lentivirus particles encoding C1C-EGFP and the indicated nanobody. 24 h post transduction, gene expression was induced with dox and 24 h later, cells were treated with NLRC4 activator MxiH as in C and E. 1 h post treatment, cells were harvested, fixed, and analyzed by flow cytometry to determine the fraction of C1C-EGFP$^+$ and thus VHH-expressing cells (**I**), total cell count over 30 s (**J**), and the fraction of C1C-EGFP$^+$ cells assembling ASC specks (**K**). **L** THP-1 cell lines expressing C1C-mCherry (dox-inducible) as well as the indicated VHH-EGFP fusions (constitutively) were differentiated with PMA, treated with dox for 24 h, and subjected to stimulation with NLRC4 agonist MxiH as in C and E, in presence of 40 µM VX. Cells were harvested and ASC specks were quantified by flow cytometry. **M, N** PMA-differentiated THP-1 macrophages expressing the indicated HA-tagged nanobodies were stimulated with MxiH for 1 h. Cells were lysed in SDS-PAGE buffer with 100 mM DTT (**M**) or no reducing agent (**N**), and subjected to SDS-PAGE and immunoblot with GSDMD and GAPDH antibodies. Representative immunoblots of at least three independent experiments are displayed (**M, N**). Data represent average values (with individual data points) from three independent experiments or donors ± SEM, unless mentioned otherwise. NS, not significant; **$P < 0.01$, and ****$P < 0.0001$ (unpaired two-tailed Student's $t$-test).

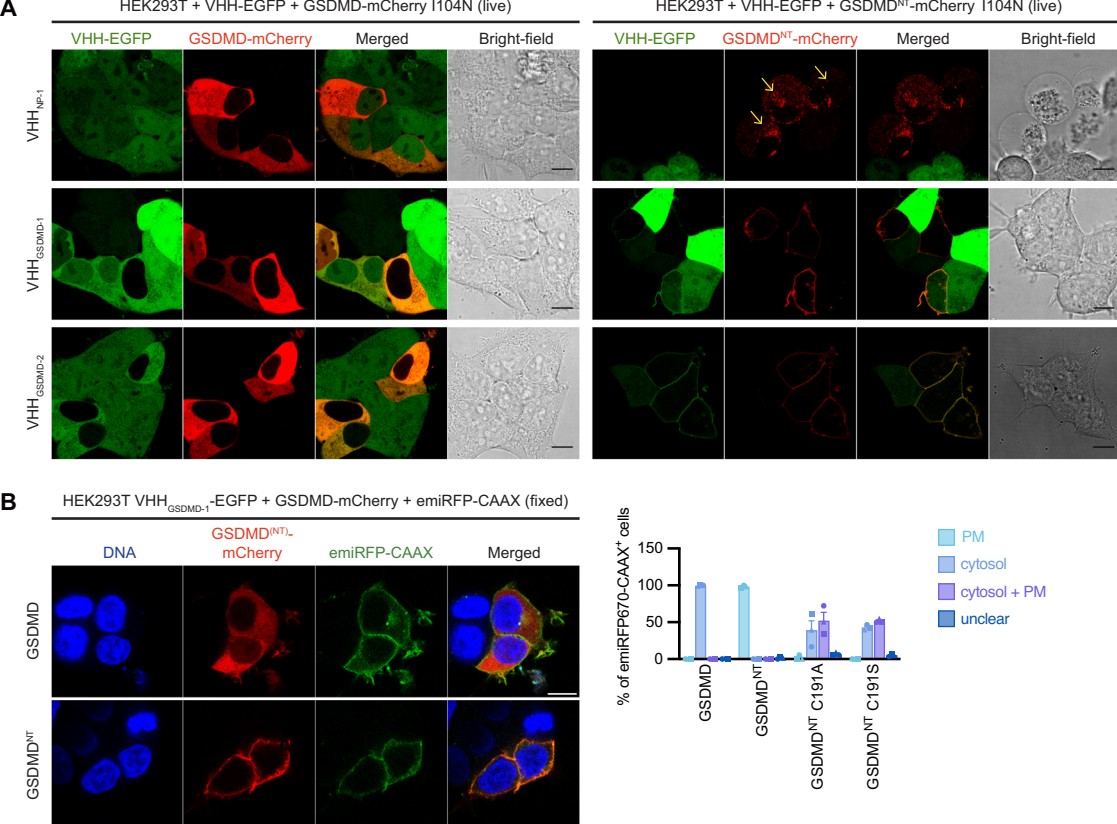

**Fig. 3 | Nanobodies preventing oligomerization still allow membrane localization of overexpressed GSDMD$^{NT}$. A** HEK293T cells stably expressing the indicated VHH-EGFP fusions were transfected with expression vectors for GSDMD-mCherry I104N (left) or GSDMD$^{NT}$-mCherry I104N (right) and analyzed by live-cell confocal imaging. Data representative of three independent experiments are shown. Yellow arrows indicate GSDMD$^{NT}$-mCherry in intracellular vesicular structures. **B** HEK293T cells stably expressing VHH$_{GSDMD-1}$-EGFP were transfected with

expression vectors for the plasma membrane marker emiRFP670-CAAX as well as the indicated GSDMD variants fused to mCherry. Representative images are shown on the left. The distribution of the indicated GSDMD variants in cells positive for emiRFP670 and mCherry were enumerated and average values from $n = 3$ independent experiments with at least $n = 30$ cells per condition are displayed ± SEM on the right. PM, plasma membrane. Scale bars, 10 µm.

pyroptotic cells, while green fluorescence started to accumulate in intracellular sites, co-localizing with CMO (yellow arrows). Z stacks of treated cells demonstrated that the co-localization of GSDMD$^{NT}$-mNG and CMO occurred in the middle of round-up, but intact cells (Figure S4B). Lastly, we could confirm that the antagonistic nanobodies

can also stabilize endogenous GSDMD$^{NT}$ in the plasma membrane. We fixed cells expressing nanobodies after stimulation, stained the plasma membrane with wheat germ agglutinin (WGA), permeabilized cellular membranes, and finally stained with an antibody that specifically recognizes processed GSDMD$^{NT}$ (Figs. 4E, F). Anti-GSDMD$^{NT}$ staining

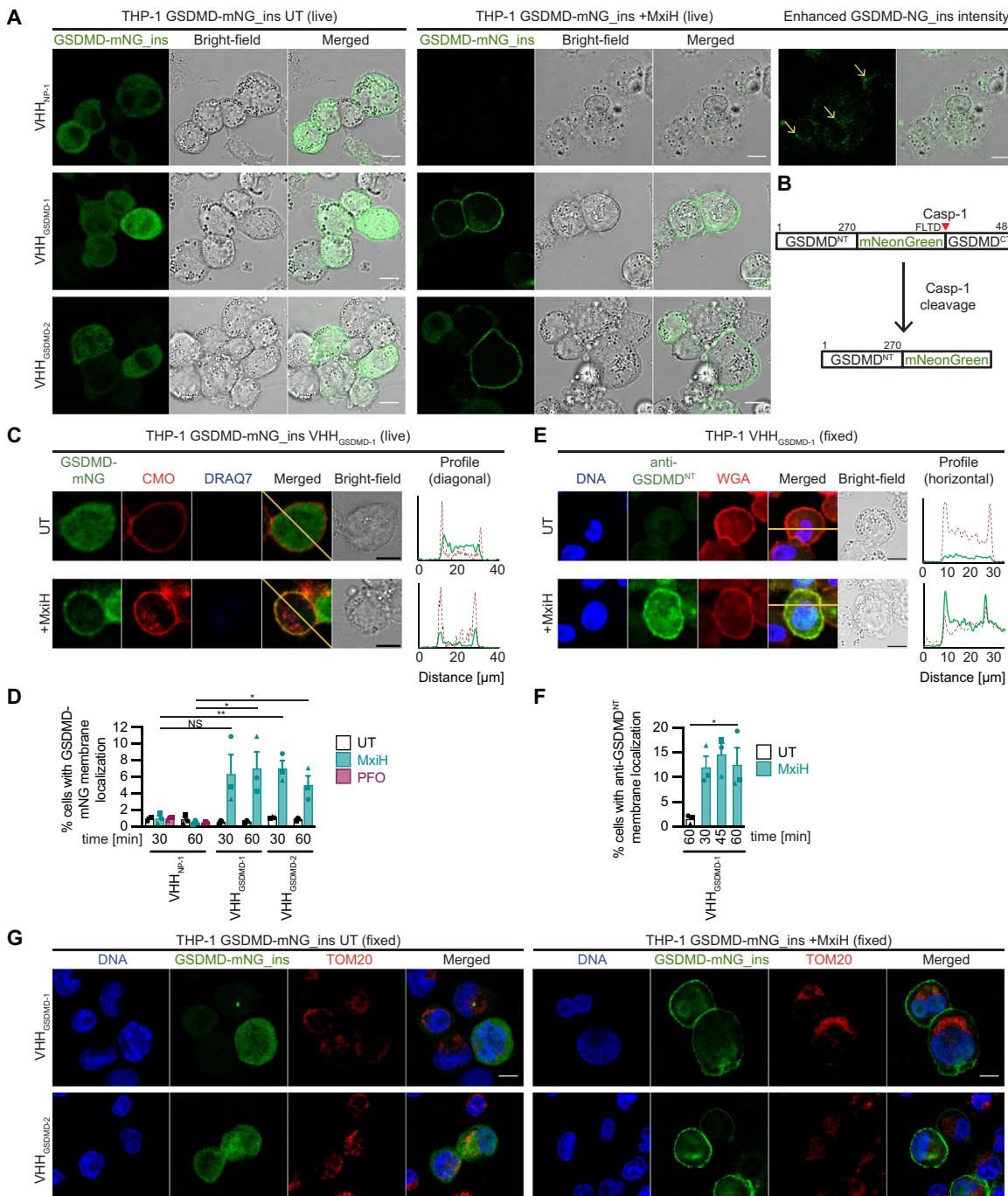

**Fig. 4 | Nanobodies preventing oligomerization still allow membrane localization of processed GSDMD^NT.** **A–D** PMA-differentiated THP-1 cells expressing GSDMD-mNeonGreen_ins (GSDMD-mNG_ins) and the indicated HA-tagged nanobodies were stimulated with MxiH for 60 (**A**) or 30 and 60 (**C**, **D**) minutes as described in Fig. 2C. A schematic representation of GSDMD-mNG_ins before and after cleavage by caspase-1 is displayed in (**B**). The plasma membrane of cells in (**C**) and (**D**) was stained with CellMask Orange (CMO) prior to stimulation. Cells were recorded by live cell confocal microscopy and representative images are displayed (**A**, **C**). Fluorescence intensities along the orange diagonal are displayed to indicate plasma membrane localization of GSDMD^NT-mNG (**C**). The fraction of cells with clear plasma membrane localization of GSDMD^NT-mNG was enumerated and average values from $n = 3$ independent experiments with at least $n = 330$ cells per condition are displayed ± SEM (**D**). **E**, **F** PMA-differentiated THP-1 cells expressing VHH_GSDMD-1-HA were stimulated with MxiH for 15, 30, and 60 minutes. Cells were

fixed, stained with fluorescent wheat germ agglutinin (WGA), fixed again, permeabilized, and stained for cleaved GSDMD^NT (anti-GSDMD^NT) and DNA (Hoechst 33342). Cells were recorded by confocal microscopy and representative images are displayed (**E**). Fluorescence intensities along the orange diagonal are displayed to indicate plasma membrane localization of GSDMD^NT. The fraction of cells with clear plasma membrane localization of GSDMD^NT was enumerated and average values from $n = 3$ independent experiments with at least $n = 100$ cells per condition are displayed ± SEM (**F**). **G** PMA-differentiated THP-1 cells expressing GSDMD-mNG_ins and the indicated HA-tagged nanobodies were stimulated with MxiH, fixed, and stained for DNA (Hoechst 33342) and mitochondria (anti-TOM20). Representative confocal microscopy images of at least three independent repeats are displayed. Scale bars, 10 μm. NS, not significant; *$P < 0.05$, and **$P < 0.01$ (unpaired two-tailed Student's *t*-test).

was only detected after inflammasome activation in the presence of antagonistic nanobodies and was localized to the plasma membrane. Taken together, our data confirmed that GSDMD cleavage releases GSDMD[NT], which inserts into the plasma membrane as a monomer before pores are formed by oligomerization. Importantly, when GSDMD[NT] pore formation was inhibited, GSDMD[NT] was not detected in defined intracellular structures and in particular, no co-localization with mitochondria was apparent (Fig. 4G). As monomeric GSDMD[NT] can directly insert into the plasma membrane and builds up the pore monomer by monomer, there does not seem to be a need to oligomerize prepores before membrane insertion.

## Inhibition of pore formation by antagonistic GSDMD nanobodies augments caspase-1 activity and triggers caspase-1-dependent apoptosis

When analyzing THP-1 macrophages expressing different VHH-EGFP fusions in combination with the C1C-mCherry inflammasome reporter with live cell microscopy, we observed inflammasome assembly in presence of VHH[GSDMD-1] and VHH[GSDMD-2] as indicated by ASC speck formation (Fig. 5A). Interestingly, cells with ASC specks exhibited blebs or were fragmented into multiple membrane-surrounded structures – morphologies more typically associated with apoptosis and apoptotic bodies (also see Fig. 2G). Cells with ASC specks expressing the VHH[NP-1] control, however, were round-up with a 'balloon-like' morphology as expected for cells undergoing pyroptosis (Fig. 5A, Fig. 2G). We quantified the observed morphologies and found that only cells expressing no or control nanobodies exhibited pyroptotic morphology and took up the DNA dye SYTOX Green upon inflammasome activation (Figs. 5B, C). Cells expressing antagonistic GSDMD nanobodies, in contrast, did not take up DNA dyes and exhibited apoptotic morphology (Fig. 5B). Similar apoptotic morphologies could be observed for the primary human macrophages transduced with antagonistic GSDMD nanobodies and C1C-EGFP as described above (Fig. 5F, Fig. 2H).

We next treated THP-1 cells expressing C1C-EGFP and HA-tagged nanobodies with NLRC4 or NLRP3 agonists in the absence of caspase-1 inhibitors. When we analyzed them by flow cytometry, we only measured C1C-EGFP specks in cells expressing VHH[GSDMD-1] and VHH[GSDMD-2], but not in cells expressing control nanobodies or no nanobodies (Figure S5D, E). This confirms that pyroptotic cells were ruptured during sample processing as observed before, while apoptotic cells could be analyzed by flow cytometry.

To probe for bona fide apoptosis, we next stained the different PMA-differentiated THP-1 cell lines for cleaved caspase-3 and quantified the fraction of cells positive for cleaved caspase-3 by flow cytometry. Staurosporine treatment for 20 h was used as a positive control and resulted in more than 60% of the cells positive for cleaved caspase-3 (Figs. 5G, H). Both VHH[GSDMD-1]- and VHH[GSDMD-2]-expressing cells showed a clear population of cells positive for cleaved caspase-3 after treatment with MxiH, indicating that the apoptotic effector caspase-3 is active (Figs. 5G, H). Interestingly, caspase-3 activation seems to be caspase-1-dependent since it was strongly reduced in presence of VX. Direct activation of caspase-8 by recruitment and autoproteolytic activation on ASC specks had been reported earlier[35–37]. As caspase-3 activation was largely blocked by VX, caspase-1-independent activation of caspase-8 does not seem to have a major contribution to caspase-3 activation. Yet the residual fraction of cells positive for caspase-3 cleavage after VX treatment may result from direct activation of caspase-8 on ASC specks, as we no longer observed caspase-3 activation in cells expressing VHH[ASC] or in THP-1 ΔASC cells (Fig. 5I,J). While NLRC4[CARD] had been reported to directly recruit caspase-1[CARD] in the absence of ASC[38], no LDH release was observed in THP-1 ΔASC cells treated with MxiH, indicating that ASC-independent caspase-1 activation did not contribute in our experimental conditions (Fig. 6A).

To measure caspase-3 activity independent of cell death or rupture, we performed caspase Glo-assays to measure the activity of caspase-3/7 in THP-1 macrophages (Figure S5G). Here, caspase activity is determined in lysates derived from the cells and the supernatant, whereby the caspase-3-specific peptide DEVD is cleaved to render a substrate available to luciferase. Strong caspase-3/7 activity was observed in MxiH-treated cells expressing VHH[GSDMD-1] or VHH[GSDMD-2], but not in cells expressing control nanobodies. Again, this activity was completely dependent on ASC.

Analysis of THP-1 macrophage cell lysates by immunoblot confirmed cleavage of caspase-3, caspase-7, and the caspase-3 substrates PARP and GSDME specifically in those samples with caspase-3 activity, i.e., in cells in which antagonistic GSDMD nanobodies prevented pore formation (Figure S5A). Remarkably, cleavage of GSDME in these cells does not seem to be sufficient to assemble functional GSDME pores, as we did not observe pyroptosis, IL-1β release, or DRAQ7 uptake (Fig. 2G, Figure S2B). GSDME therefore does not seem to play a major role in the death of VHH[GSDMD]-expressing cells. The GSDMD fragment GSDMD[p20] in cells expressing VHH[GSDMD-1] (Fig. 2M, Figure S5B) also coincides with enhanced caspase-3 activation and disappears upon addition of a caspase-3/7 inhibitor, suggesting that it represents GSDMD[NT] cleaved by caspase-3 (Figure S5B). Of note, GSDMD[p20] is only observed in cell lines expressing VHH[GSDMD-1] but not VHH[GSDMD-2], perhaps because access of caspase-3 is occluded by VHH[GSDMD-2].

Lysates of THP-1 macrophages expressing VHH[GSDMD-1] and VHH[GSDMD-2] not only contained cleaved caspase-3, but also processed caspase-8, processed caspase-9, and cleaved tBID, as well as detectable caspase-8 activity (Figure S5C,D, H), indicating the activation of both the intrinsic and extrinsic apoptosis pathway, or the activation of feedback mechanisms involving the upstream caspases. Not only caspase-3 activity, but also caspase-8 and caspase-9 activity as well as tBID cleavage are dependent on caspase-1 and ASC, suggesting that caspase-1 activated at the inflammasome seems to be the key regulator of the alternative cell death program (Figures S5E, F, H). Only for caspase-8 there is some residual processing that can also be seen in the absence of GSDMD VHHs and the presence of caspase-1 inhibitors (Figure S5E). This caspase-8 activation is completely ASC dependent since it is absent in the THP-1 ΔASC cells (Figure S5F), suggesting that a small portion of the caspase-8 is cleaved at the ASC speck, independent of caspase-1 as concluded above[35–37].

We next quantified the caspase-1 activity of THP-1 macrophages upon MxiH stimulation using caspase-1 Glo assays. Surprisingly, we found that caspase-1 activity was increased up to 6-fold in presence of VHH[GSDMD-1] or VHH[GSDMD-2] compared to the pyroptotic cells expressing VHH[NP-1] (Fig. 6B). This is remarkable, as the assembly of ASC specks was comparable in all samples (Fig. 2L). We therefore hypothesize that the ability to form GSDMD pores has a profound impact on caspase-1 activity, suggesting GSDMD pores downregulate caspase-1 activity in a so far elusive mechanism. Caspase-1 ultimately serves as the master regulator for downstream cell death, as only the enhanced caspase-1 activity observed in the absence of GSDMD pores was sufficient to activate caspase-3 and apoptosis.

The relatively low caspase-1 activity in cells undergoing pyroptosis may be a consequence of ion fluxes and/or the release of caspase-1 into the supernatant through GSDMD pores, even though caspase-1 activity was measured in samples derived from the cells and the supernatant. The altered environment may for example compromise the stability of short-lived active caspase-1. To test the impact of pores on caspase-1 activity, we treated THP-1 macrophages expressing VHH[GSDMD-1] or VHH[GSDMD-2] with MxiH in the presence of the pore-forming toxin PFO. PFO forms pores with a diameter of 25-30 nm, i.e., a similar if not slightly larger diameter than GSDMD pores[39]. To avoid additional activation of NLRP3 by potassium efflux through PFO pores, NLRP3 was activated in the presence of NLRP3 inhibitor CRID3. PFO-induced

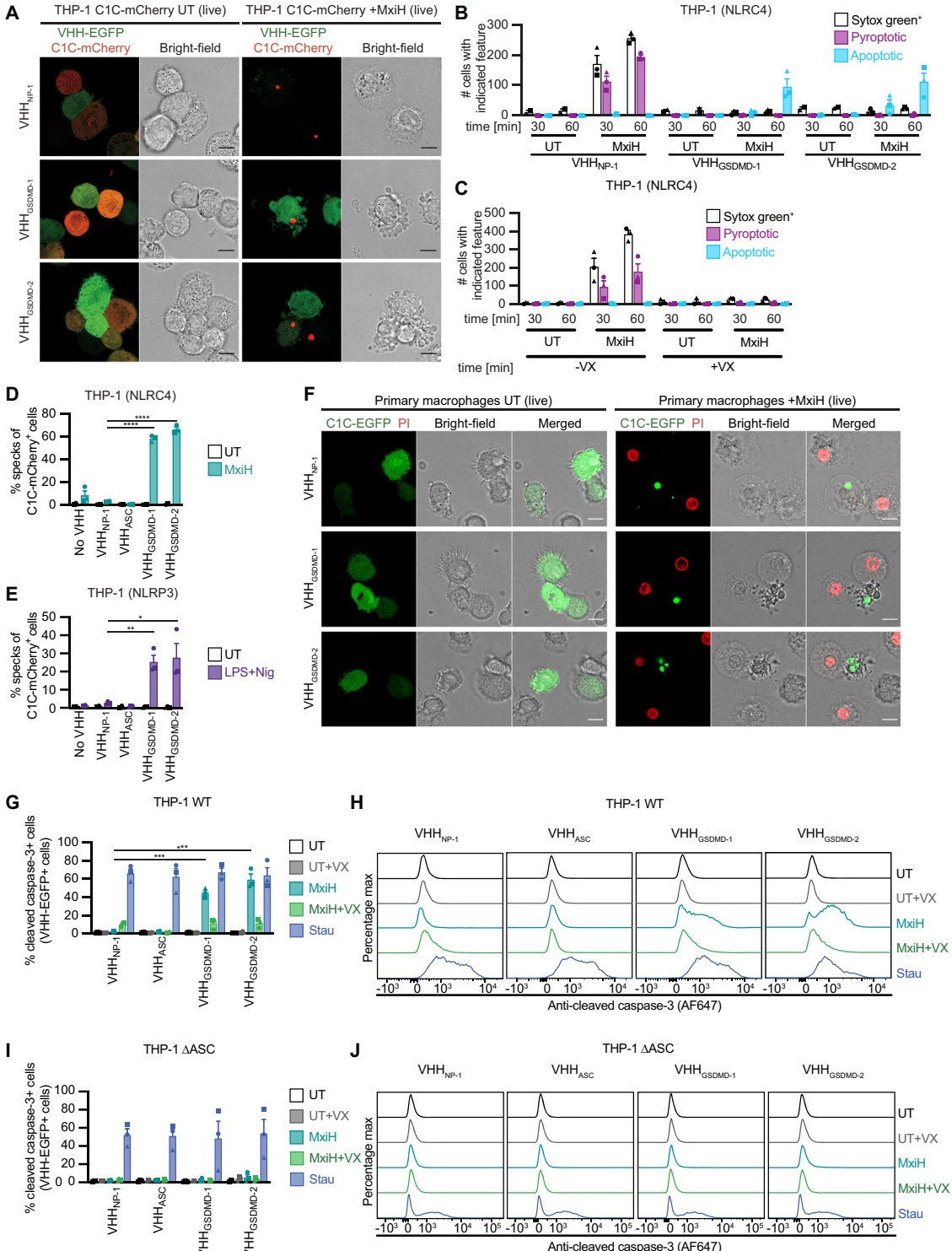

pore formation was confirmed by influx of DRAQ7 (Fig. 6E). We indeed observed a dose-dependent reduction in caspase-1 activity upon higher levels of PFO administration, indicating that the formation of pores in the cellular membrane is sufficient to reduce the activity of caspase-1 (Figs. 6C, D).

In summary, we propose that the observed apoptosis in the absence of functional GSDMD pores is completely dependent on inflammasome assembly. The augmented caspase-1 activity observed in the absence of GSDMD pores seems to be central to process apoptotic initiator and effector caspases.

## Recombinant antagonistic GSDMD nanobodies inhibit pyroptosis when administered extracellularly

GSDMD is linked to a growing list of diseases and thus an eminent drug target[23–25], although the development of specific GSDMD inhibitors was not successful to date[40–43]. We therefore tested if uptake of recombinant nanobodies through GSDMD pores, as we had observed for VHH_ASC[28], was able to counteract inflammation. We added increasing concentrations of purified nanobodies to the culture medium of THP-1 macrophages treated with MxiH. The highest concentrations of VHH_GSDMD-1 and VHH_GSDMD-2 reduced LDH release to

**Fig. 5 | Inhibition of pore formation by antagonistic GSDMD nanobodies triggers caspase-1-dependent apoptosis. A, D–E, G–J** THP-1 cell lines expressing C1C-mCherry (dox-inducible) as well as the indicated VHH-EGFP fusions (constitutively) were differentiated with PMA, treated with dox for 24 h, and stimulated with NLRC4 agonist MxiH for 1 h as described in Fig. 2C (**A, D, G–J**), with NLRP3 agonist LPS and Nig as described in Fig. 2D (**E**), or with 5 μM staurosporine (Stau) for 20 h to trigger apoptosis (**G–J**). Stimulation was performed in the absence or presence of VX as indicated. **A** Cells were recorded by live cell confocal microscopy and images representative of three independent experiments are displayed. Scale bars, 10 μm. **D, E** Cells were harvested and analyzed by flow cytometry to quantify C1C-mCherry specks. **G–J** Cells were harvested, stained with antibodies specific for cleaved caspase-3 and Alexa Fluor 647 (AF647)-coupled secondary antibodies, and the fraction of cells positive for cleaved caspase-3 was quantified by flow cytometry (**G, I**) Representative histograms of cell lines with the indicated treatments are

presented in (**H, J**). **B, C** WT THP-1 cells (**C**) or THP-1 cell lines expressing the indicated HA-tagged VHHs (**B**) were differentiated with PMA, and stimulated with NLRC4 agonist MxiH as described in Fig. 2C, but in the presence of 100 nM SYTOX Green nucleic acid stain. Cells were recorded by live cell confocal microscopy including bright field recordings. The absolute number of cells positive for SYTOX Green as well as cells with pyroptotic and apoptotic morphology were enumerated per tile scan (675 μm × 675 μm) and average values from three independent experiments ± SEM are displayed. **F** Primary GM-CSF-differentiated monocyte-derived human macrophages were transduced and stimulated as described in Figs. 2H–K. Cells were recorded by live-cell confocal microscopy and images representative of three independent donors are displayed. Scale bar, 10 μm. Data represent average values (with individual data points) from three independent experiments ± SEM. *P < 0.05, **P < 0.01, ***P < 0.001, and ****P < 0.0001 (unpaired two-tailed Student's t-test).

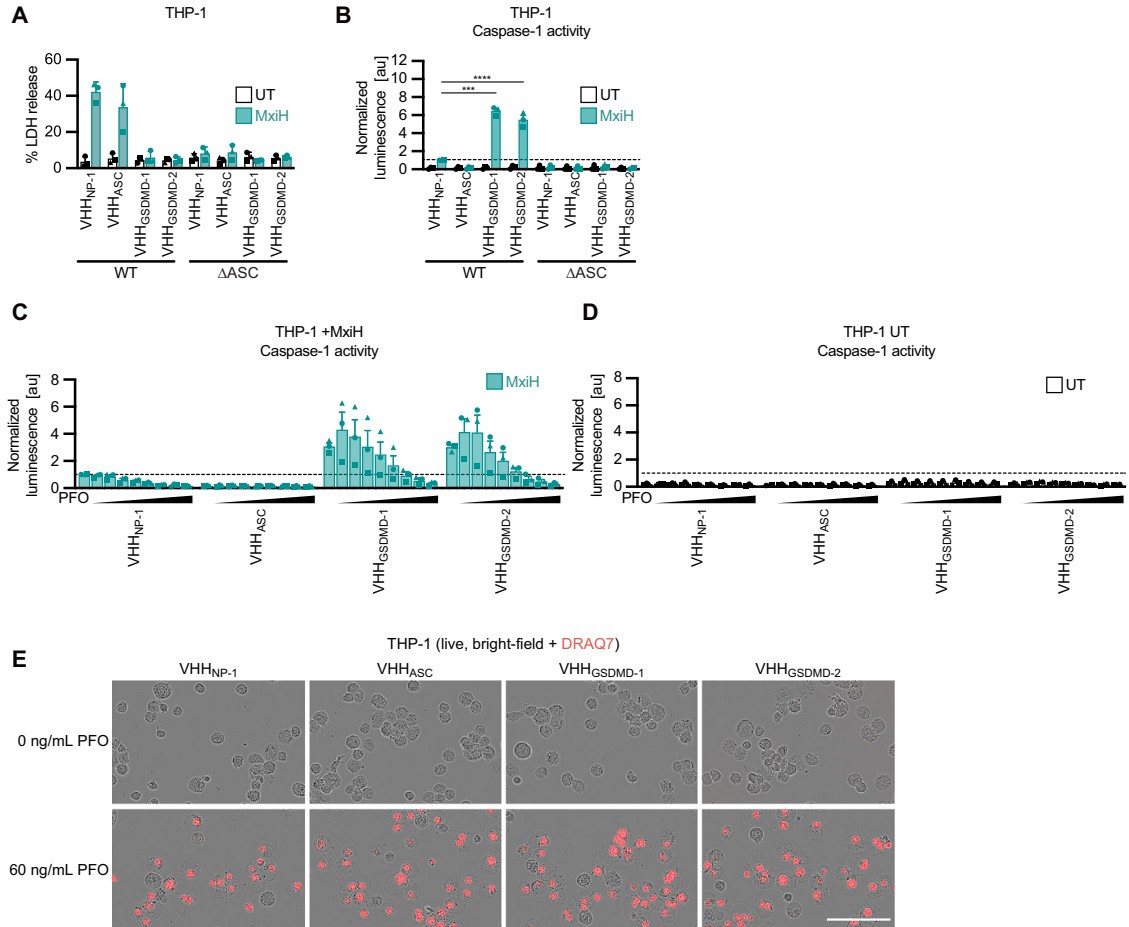

**Fig. 6 | Inhibition of pore formation by antagonistic GSDMD nanobodies augments caspase-1 activity.** PMA-differentiated THP-1 WT (**A–E**) or THP-1 ΔASC (**A, B**) cells constitutively expressing the indicated HA-tagged nanobodies were stimulated with MxiH for 1 h as described in Fig. 2C. Where indicated, experiments were performed in the presence of increasing PFO concentrations (0, 5, 10, 20, 30, 60, 120, 240, and 480 ng/mL) (**C, D**), or 60 ng/mL PFO (**E**). **A** LDH release was measured and normalized to cells lysed in Triton X-100. **B–D** Cells and supernatants were harvested to measure caspase-1 activity using Caspase-Glo assays. Activity was

corrected for cell numbers per sample using CTB values and normalized to MxiH-treated cells expressing VHH_NP-1 (indicated as dashed line). **E** THP-1 cells treated in the presence of DRAQ7 were recorded with an Incucyte Live-Cell Imaging system. Representative image of cells left untreated (0 ng/mL PFO) or treated with 60 ng/mL PFO after 30 minutes are displayed. Scale bar, 100 μm. Data represent average values (with individual data points) from three independent experiments ± SEM. ***P < 0.001, and ****P < 0.0001 (unpaired two-tailed Student's t-test).

background levels (Fig. 7A) and the secretion of IL-1β was also substantially reduced (Fig. 7B). A similar reduction in LDH release was observed after stimulation of NLRP3 inflammasomes (Fig. 7C), although the reduction of cytokine release was only partial (Fig. 7D). Cytosolic expression of VHH_GSDMD-1 reduced pyroptosis triggered by overexpressed murine GSDMD^NT (Figure S6A), indicating some cross-reactivity, despite the negative LUMIER data. Yet, no inhibition of LDH

release was observed in murine macrophages treated with extracellular GSDMD nanobodies upon stimulation (Figure S6B), precluding animal experiments in mice. We next sought to quantify survival or delayed cell death of nanobody-treated cells using a readout independent of plasma membrane integrity. We found that the reducing potential of cells as measured by CellTiter-Blue (CTB) assays was abrogated after NLRC4 inflammasome assembly, whereas addition of

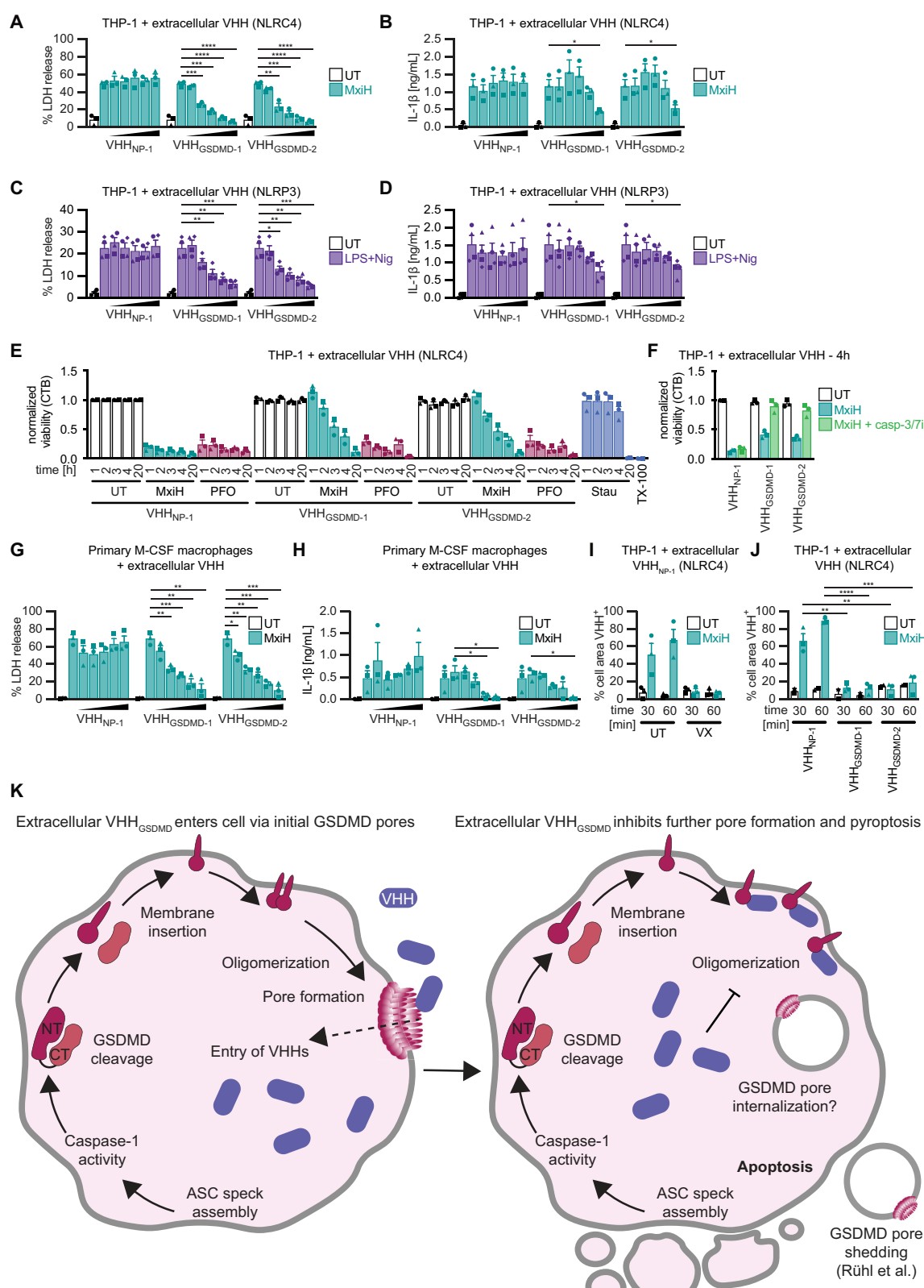

antagonistic GSDMD nanobodies completely inhibited loss of reducing potential after 1 h (Fig. 7E). Over time, however, cells lost reducing potential, indicating that cell death was delayed, but not completely abrogated. 20 h post treatment, the CTB signal was completely lost. Reducing capacity could be fully rescued by caspase-3/7 inhibition, confirming that the delayed cell death in presence of GSDMD

nanobodies occurs by apoptosis (Fig. 7F). Extracellular administration of the antagonistic GSDMD nanobodies thus changed the inflammasome-mediated type of cell death from pro-inflammatory pyroptosis to non-inflammatory apoptosis (Figure S6C). To validate the inhibitory effects of extracellular nanobodies in a physiologically relevant in vitro model, we repeated the same experiments in primary

**Fig. 7 | Recombinant antagonistic GSDMD nanobodies inhibit pyroptosis when administered extracellularly. A**–**D** PMA-differentiated THP-1 cells were treated with MxiH as described in Fig. 2C (**A**, **B**) or with LPS and Nig. as described in Fig. 2D (**C**, **D**) in the presence of increasing concentrations (2, 20, 50, 100, and 200 µg/mL) of the indicated recombinant nanobodies. **A**, **C** LDH release was measured and normalized to cells lysed in Triton X-100. **B**, **D** IL-1β in the supernatant was quantified by HTRF. **E**, **F** PMA-differentiated THP-1 cells were treated with MxiH, 100 ng/mL PFO, or 5 µM Stau for 1, 2, 3, 4, and 20 h in the presence of 200 µg/mL of the indicated nanobodies. The reducing capacity as a readout for viability was determined by CellTiter-Blue (CTB) assay and normalized to untreated cells in the presence of VHH$_{NP-1}$. As a positive control, cells were incubated with 1% Triton X-100. Where indicated, cells were treated for 4 h in the presence of 40 µM caspase-3/7 inhibitor (casp-3/7i) (**F**). **G**, **H** M-CSF-differentiated primary human macrophages from independent donors were treated as in Fig. 7, A and B, and LDH release (**G**) and IL-1β secretion (**H**) were quantified as before. **I**, **J** THP-1 cells (**I**) or THP-1 cell lines expressing the indicated HA-tagged VHHs (**J**) were differentiated with PMA, labeled with CMO, and stimulated with MxiH in the presence of 200 µg/mL VHH$_{NP-1}$ total (60 µg/mL of the nanobody was AF647 labeled). Where indicated, stimulation was performed in the presence of 40 µM VX (**I**). Cells were recorded by live cell confocal microscopy including bright field recordings. Cell areas (mostly containing a single cell) were identified using the CMO staining by Imaris. Cell areas were scored as VHH-positive (VHH$^+$) when VHH$_{NP-1}$-AF647 intensity was at least 80 (corresponding to ca. 75% of the mean intensity outside the cells in the first data set). The fraction of VHH$^+$ cell areas was normalized to the total cell area. Average values from three independent experiments ± SEM are displayed. **K** Model for the inhibition of pyroptosis by antagonistic GSDMD nanobodies added to the extracellular space. Inflammasome-dependent cleavage of GSDMD allows insertion of monomeric GSDMD$^{NT}$ into the plasma membrane, where monomers oligomerize and assemble pores that penetrate the membrane. This allows influx of extracellular nanobodies into the cytosol (left). These prevent the formation of new GSDMD pores by stabilizing monomers. Existing GSDMD pores are removed by membrane repair mechanisms, including shedding as well as potentially endocytosis, which ultimately prevents cell death by pyroptosis (right). Cells bearing conventional inflammasomes will eventually die by non-inflammatory apoptosis. Data on LDH, IL-1β release, and CTB assays represent average values (with individual data points) from three independent experiments or donors ± SEM. *$P < 0.05$, **$P < 0.01$, ***$P < 0.001$, and ****$P < 0.0001$ (unpaired two-tailed Student's $t$-test).

human M-CSF macrophages. Here, LDH release was similarly inhibited in a dose-dependent manner and IL-1β release was completely abrogated at the highest concentrations (Figs. 7G, H).

Using Alexa Fluor 647-labeled VHH$_{NP-1}$ (Figs. 7I, J, Figure S6D, E), we found that uptake of fluorescent nanobodies by pyroptotic cells required inflammasome assembly, caspase-1 activity, and GSDMD pore formation, while only minute amounts of nanobody were taken up by endocytosis. We hypothesize that nanobodies enter cells with inflammasomes upon formation of the first GSDMD pores, before the lytic stage of pyroptosis (Fig. 7K). Cytosolic nanobodies may thus prevent any further GSDMD pore assembly, which seems to be sufficient to prevent cell lysis. We next tested whether the transient pores that formed in the presence of antagonistic GSDMD nanobodies were indeed sufficient to allow an influx of molecules into the cytosol. We thus conducted wide-field microscopy experiments with the Incucyte system and found that THP-1 cells treated with NLRC4 activators in the presence of GSDMD nanobodies took up modest levels of DNA dye SYTOX Green, although uptake per cell was lower than in (pyroptotic) cells in the presence of control nanobodies (Figure S7A-D). This indicated that GSDMD pores do form in the presence of inhibitory GSDMD nanobodies, but that influx of molecule is substantially reduced, likely because GSDMD pores are not sustained. To sensitively detect uptake of minute amounts of fluorescent nanobodies through transient GSDMD pores, we produced fluorescently labeled VHH$_{ASC}$, which we expected to enrich on ASC specks. We treated THP-1$^{C1C-EGFP}$ cells with NLRC4 activators in the presence of nanobodies, and spiked in low concentrations of VHH$_{ASC}$ AF647, which are not expected to affect inflammasomes. Inflammasome assembly was detected by recruitment of C1C-EGFP to ASC specks (Figure S8, A-C). Substantial cellular uptake of fluorescent VHH$_{ASC}$ (indicated by co-localization with nuclear dyes) was only observed in pyroptotic cells in the presence of control nanobody VHH$_{NP-1}$ (Figure S8D). Yet, we observed enrichment of VHH$_{ASC}$ on ASC specks in the presence of both VHH$_{GSDMD-1}$ and VHH$_{NP-1}$, and this was abrogated by caspase-1 inhibition (Figure S8E). This proved that nanobodies indeed enter sublytic cells in a caspase-1-dependent manner, likely through transient GSDMD pores. Using confocal live cell imaging of individual cells, we found that uptake of fluorescently labeled nanobodies in the presence of VHH$_{GSDMD-1}$ preceded apoptosis (Figure S9, S10 and movies S3-5). In contrast to cells undergoing pyroptosis in the presence of control nanobodies (Figure S11, movie S6), apoptotic cells did not take up substantial amounts of the DNA dye PI (our confocal microscopy setting did not detect sublytic uptake of DNA dyes). Based on our experiments with fluorescent GSDMD$^{NT}$ fusions, it is likely that the early GSDMD pores are rapidly removed by membrane repair processes. Initial pore formation

may well explain the remaining IL-1β secretion since the cytokine could still be released through early sublytic GSDMD pores. Altogether, these results show that the nanobodies are potent inhibitors of inflammasome-induced pyroptosis when administered extracellularly, which reveals their interesting therapeutic potential. Importantly, early GSDMD pore formation does not seem to be a terminal event, as cells could still be rescued from pro-inflammatory pyroptosis by antagonistic GSDMD nanobodies.

## Discussion

GSDMD pore formation is the effector mechanism that mediates cell death by pyroptosis as well as the secretion of mature IL-1β and IL-18. Despite the detailed structural understanding of GSDMD pores, critical molecular aspects of pore formation remained unknown as the process cannot be easily studied in relevant cells, primarily because pyroptotic cells do not weather sample preparation for microscopy and flow cytometry. In this study, we discovered two GSDMD-targeting nanobodies, which potently inhibit pyroptosis by preventing the oligomerization of GSDMD$^{NT}$, thereby stabilizing monomeric GSDMD$^{NT}$. In a parallel study, Kopp et al. solved the crystal structure of VHH$_{GSDMD-2}$ and VHH$_{GSDMD-6}$ in a complex with full-length GSDMD[44]. This confirms that VHH$_{GSDMD-2}$ binds to an epitope of GSDMD that forms the oligomerization interface in GSDMD$^{NT}$ pores. Importantly, we observed that monomeric GSDMD$^{NT}$ could still localize to the plasma membrane. As the observed clear partitioning into the plasma membrane was dependent on amino acids required for membrane insertion, we concluded that nanobody-bound GSDMD$^{NT}$ inserts into the plasma membrane. This implies that cleavage of GSDMD is sufficient to mediate all steps necessary for membrane insertion, including the electrostatically driven membrane association and the conformational changes that likely expose the extended beta-sheet that dips into the plasma membrane. This for the first time allowed us to observe and study GSDMD membrane recruitment in (living) human cells, supporting the conclusion that pores can grow monomer by monomer in a target membrane. These results are in line with previous in vitro findings in artificial membranes, showing that human GSDMD$^{NT}$ assembles smaller arcs or slits, which grow into ring-shaped assemblies, although monomeric GSDMD$^{NT}$ could not be detected by atomic force microscopy[4,15]. Likewise, atomistic molecular dynamics simulations predicted that small GSDMD$^{NT}$ assemblies can already form ion-conducting membrane pores and provide a plausible pathway to pore opening in intact bilayers[45]. Earlier work had proposed that GSDMA3$^{NT}$ or GSDMD$^{NT}$ form ring-like prepores associated with membranes, in which the N-terminal domains maintain a globular conformation as found in full-length gasdermin[13,14]. This model implied that a coordinated conformational change in all subunits gives rise to the eventual β-barrel structure

that is inserted in the cell membrane[13,14]. Our data suggests that monomers of GSDMD[NT] undergo conformational changes that allow membrane insertion, even if oligomerization is prohibited with nanobodies. This indicates that the assembly of prepores is not necessary for membrane insertion and therefore unlikely to be critical, although it does not completely rule out that two different pathways to pore-formation exist in parallel[16]. It is in principle conceivable that free GSDMD[NT] behaves differently from nanobody-bound GSDMD[NT]. While every experimental perturbation may affect the analyzed molecules, our conclusions are derived from direct observation of (endogenous) GSDMD in relevant cell types, in which we observed membrane insertion at similar time points as pore formation in the unperturbed situation. Alternative models for pore formation, however, were derived from in vitro experiments with recombinant proteins reconstituted in artificial membranes[13,14] and did not directly visualize GSDMD[NT] in cells. Moreover, binding of VHH[GSDMD-2] does not change the GSDMD structure[44], rendering it very unlikely that binding of the nanobody unlocks the insertion-competent conformation of GSDMD[NT], while free GSDMD[NT] would stay in the globular conformation that it assumes in full-length GSDMD. Interestingly, GSDMD[NT] almost exclusively inserts into the plasma membrane if oligomerization is inhibited by binding of antagonistic nanobodies. This demonstrates that the plasma membrane is indeed the primary target of GSDMD[NT] pores and mitochondrial localization of GSDMD[NT] is unlikely during the first phase of pore formation in macrophages.

The antagonistic GSDMD nanobodies described in this study also provide new insights into the interconnectivity of the different cell death pathways in macrophages. In the presence of fully cleaved endogenous GSDMD in cells expressing VHH[GSDMD-1] or VHH[GSDMD-2], macrophages undergo apoptosis that is dependent on inflammasome assembly, ASC specks, and caspase-1 activity. Inflammasome-mediated apoptosis has previously been reported in caspase-1 knockout cells, in cells expressing catalytically inactive caspase-1, as well as in cells lacking GSDMD[1,36,37,46,47]. Caspase-1-mediated apoptosis in the absence of GSDMD pores is mechanistically different from the described ASC-dependent caspase-8-mediated apoptosis, which was observed in the explicit absence of caspase-1 activity[36,37,47]. Although ASC[PYD] can nucleate polymerization of caspase-8 death effector domains (DEDs) in absence of caspase-1[35], we found that caspase-3 activation was minimal in the absence of caspase-1, suggesting that direct recruitment of caspase-8 to ASC specks does not substantially contribute to the observed early apoptosis. Interestingly, apoptosis observed in our system followed a kinetic comparable to pyroptosis, with MxiH-stimulated cells already exhibiting caspase-3 activity as well as apoptotic morphology within 20-60 minutes after treatment. In contrast, canonical apoptosis, e.g., triggered by staurosporine, is a slower process in which caspase-3 activity and cell death are only emerging after more than three hours[48] (see also Fig. 7E).

Importantly, we report that inflammasome activation in the absence of GSDMD pore formation strongly augments caspase-1 activity. Only this enhanced activity resulted in efficient cleavage of caspase-3, caspase-7, and their substrates. We therefore propose a key regulatory role for the caspase-1 activity, which seems to be reduced when pores are formed. It is possible that the most active form of cleaved caspase-1, the $(p33/p10)_2$ form[49,50], is stabilized in the absence of pores by preventing or delaying the secondary cleavage between the CARD and p20, which is associated with loss of activity. Pore formation may also provide some unidentified feedback signal to caspase-1 to dampen activity.

Remarkably, GSDME was efficiently cleaved in cells that assembled inflammasomes in the absence of GSDMD pores. Yet, we did not observe pyroptosis mediated by GSDME[NT]. This corroborates earlier findings that suggested that GSDME-induced lytic cell death does not play a major role in macrophages[51–53]. In contrast, overexpressed GSDME[NT] in HEK293T cells (see Fig. 2B)[54,55] as well as GSDME cleaved by caspase-3 in SH-SY5Y and MeWo cells[56] are sufficient to initiate

pyroptosis. This suggests that GSDME[NT] may be subject to additional layers of regulation.

Our proof-of-concept experiments lastly highlight the interesting therapeutic potential of antagonistic nanobodies VHH[GSDMD-1] and VHH[GSDMD-2]. Targeting GSDMD would not only prevent inflammation upon canonical but also non-canonical inflammasome stimuli. We could show that the extracellular addition of the nanobodies drastically reduces pyroptosis and the release of the pro-inflammatory cytokine IL-1β in both PMA-differentiated THP-1 macrophages as well as primary human macrophages. We propose that the nanobodies enter the cells upon the formation of the first sublytic GSDMD pores, rendering further GSDMD[NT] oligomerization and thus pore formation and pyroptosis impossible (Fig. 7K). One therapeutic advantage may be that the nanobodies in this scenario only target cells that have already assembled GSDMD[NT] pores, i.e., nanobodies only gain access to cells relevant for the inflammatory response.

In conclusion, we show that antagonistic GSDMD nanobodies afford informative modes of intervention by stabilizing relevant intermediates of GSDMD[NT] pore formation. The observed functional perturbation not only allowed mechanistic insights into membrane insertion and pore formation, but also provides an interesting proof of concept for the therapeutic application of recombinant nanobodies.

## Methods

### Ethics statement
Experiments with cells derived from human blood were approved by the Ethics Committee of the Medical Faculty of the University of Bonn (032/18). Alpaca immunizations were approved by the MIT Committee on Animal Care.

### Cell lines
Human embryonic kidney (HEK) 293 T cells (ATCC CRL-3216, RRID: CVCL_0063) and murine immortalized macrophages (iMacs, Latz laboratory, University of Bonn), were cultivated in DMEM GlutaMax medium (Thermo Fisher Scientific) containing 10% FBS; THP-1 cells (ATCC TIB-202, RRID: CVCL_0006) were cultured in RPMI 1640 GlutaMax medium (Thermo Fisher Scientific) containing 10% FBS and 50 μM 2-mercaptoethanol. All genetically modified cell lines were generated by lentiviral transduction using lentivirus produced with packaging vectors psPax2 and pMD2.G (kind gifts from Didier Trono, École polytechnique fédérale de Lausanne, Switzerland). All cell lines used in this study are summarized in supplementary table 1. THP-1 or HEK293T cell lines constitutively expressing VHH[GSDMD-1], VHH[GSDMD-2], VHH[GSDMD-3], VHH[NP-1], or VHH[ASC] with a C-terminal HA tag or C-terminal EGFP fusion under the control of the human elongation factor-1 α promoter (pEF1α) were generated using lentiviral vectors constructed by Gateway cloning (Thermo Fisher Scientific) using vectors modified from pRLL (a kind gift of Susan Lindquist, Whitehead Institute of Biomedical Research), followed by selection in 0.75 μg/mL puromycin (Thermo Fisher Scientific). Cell lines inducibly expressing the caspase-1[CARD]-EGFP (C1C-EGFP) or C1C-mCherry inflammasome reporter were generated using lentiviruses produced with derivates of pInducer20 (a kind gift of Stephen Elledge, Harvard Medical School)[57], followed by selection in 500 μg/mL geneticin (Thermo Fisher Scientific). C1C-EGFP is efficiently recruited to nascent ASC specks, recapitulating the recruitment of endogenous caspase-1 through its CARD. These cell lines formed the basis for further lentiviral transduction to incorporate the constitutively expressing nanobodies as described above. THP-1 ΔASC cells were generated by lentiviral transduction with derivatives of pLenti CRISPR v2 (a kind gift from Feng Zhang, Broad Institute) with the targeting sequence GCTGGATGCTCTGTACGGGA. A representative single cell clone was validated by immunoblot and genomic DNA sequencing. Derivative THP-1 ΔASC cells expressing EGFP fusions of VHH[GSDMD-1], VHH[GSDMD-2], VHH[NP-1] or VHH[ASC] were generated by lentiviral transduction with derivatives of pRRL, followed by sorting for

EGFP positive cells using a BD FacsAria Fusion cell sorter. All expression levels were verified by flow cytometry using the encoded fluorescent protein, or anti-HA staining with anti-HA B6 HA.11 (1:1000) and anti-mouse IgG Alexa Fluor 488 (1:500). Cells were fixed in 4% formaldehyde and measured using BD FACSCanto or Miltenyi MACSQuant flow cytometers. Cell lines are routinely tested for Mycoplasma contamination. All experiments involving lentiviruses were conducted in a Biosafety Level 2 laboratory.

## Primary cells

Human CD14$^+$ monocytes were isolated from human whole blood buffy coats obtained from the blood bank of the University Hospital Bonn, with consent of healthy donors and according to protocols accepted by the institutional review board of the University of Bonn. PBMCs were isolated using Ficoll-Paque PLUS (VWR) according to the manufacturer's suggestions and monocytes were purified using positive selection with paramagnetic CD14 (human) MicroBeads (Miltenyi Biotec). CD14$^+$ monocytes were differentiated into macrophages using 100 ng/mL of recombinant human M-CSF (Immunotools) or 500 U/mL of recombinant human GM-CSF (Immunotools) in RPMI 1640 GlutaMax medium supplemented with 10% FBS, 500 U/mL PenStrep, and 1 mM sodium pyruvate for 3 days. To express VHH$_{GSDMD-1}$, VHH$_{GSDMD-2}$, or VHH$_{NP-1}$ in combination with C1C-EGFP in primary macrophages, cells were lentivirally transduced. To overcome restriction by SAMHD1 in macrophages, lentivirus was produced in cells expressing a fusion protein of SIVmac251 Vpx and HIV-1 NL4.3 Vpr. Vpx-Vpr is packaged into lentivirus particles as Vpr binds to the structural protein Gag, and thus delivers Vpx into target cells, which mediates the Cullin-4a-mediated proteasomal degradation of SAMHD1[58,59]. HEK293T cells were thus transfected with psPax2, pMD2.G, pCAGGS Vpx-Vpr, and lentiviral vectors based on pInducer20bi-NA, a derivative of pInducer20-NA with the bidirectional doxycycline-inducible promoter from pTRE3G-BI (TaKaRa). Expression of VHH-HA and the C1C-EGFP inflammasome reporter was doxycycline-inducible. Lentivirus was harvested 48 h post transfection, filtered through a 0.4 μm filter, and used to transduce primary macrophages in the presence of 10 μg/mL polybrene for 6 h. The next day, expression of both the VHH and the C1C-EGFP was induced with 1 μg/mL doxycycline for 24 h.

## Plasmids

Expression vectors and lentiviral vectors described in the individual experiments were generated by Gateway and Gibson cloning. Plasmid maps and oligonucleotide sequences are shared on request.

## Proteins

**Expression and purification of His-SUMO-GSDMD, His-SUMO, His-LFn-MxiH, and PA.** Expression vectors for human His-SUMO-GSDMD and His-SUMO were generated by inserting SUMO-GSDMD or SUMO into pET28 by Gibson cloning. The His-tagged fusion of *B. anthracis* LFn (aa 1-255) and *Shigella flexneri* MxiH (LFn-MxiH) were expressed with pET-15b LFn-MxiH[25]. All proteins but PA were expressed in *Escherichia (E.) coli* LOBSTR[60] cells in Terrific Broth induced with 0.2 or 1 mM IPTG at an OD600 of 0.6. Cells were cultivated for 24 h at 18 °C and lysed by French Press or sonication with a Bandelin Sonopuls HD2070 with TT13 tip. Subsequently, the proteins were purified by Ni-NTA affinity chromatography using Ni-NTA agarose beads (Qiagen) and gel filtration with a HiLoad 16/600 Superdex 75 pg column in buffers containing 20 mM HEPES pH 7.4, 150 mM NaCl, 10% glycerol, and 1 mM DTT (His-SUMO and His-SUMO-GSDMD) or PBS (His-LFn-MxiH). To obtain unmodified GSDMD, His-SUMO was cleaved off with SUMO protease His-ULP1 and His-SUMO depleted with Ni-NTA resin. *B. anthracis* protective antigen (PA) was expressed in *E. coli* BL21(DE3) cells transformed with pGEX-6P-1 PA[25]. GST-PA was purified with Glutathione Sepharose 4B (GE Healthcare), followed by cleavage and

removal of GST with PreScission protease and Glutathione Sepharose. PA was further purified by anion exchange chromatography with a HiTrap Q HP column (GE Healthcare) and gel filtration with a HiLoad 16/600 Superdex 200 pg column (GE Healthcare) in PBS[27]. Endotoxins were removed from PA and LFn-MxiH preparations using two extractions with Triton X-114, followed by removal of remaining detergent with Bio-Beads SM-2 beads (Bio-Rad Laboratories).

**Expression and purification of nanobodies.** Coding sequences for the different GSDMD nanobodies and the control VHH$_{NP-1}$ were cloned into pHEN6-based bacterial, periplasmic expression vectors with C-terminal LPETG-His$_6$ (large scale) or HA-His$_6$ (small scale) tags using Gibson cloning. Nanobodies were expressed in *E. coli* WK6 bacteria transformed with nanobody expression vectors grown in Terrific Broth[27]. Expression was induced with 1 mM IPTG at an OD$_{600}$ of 0.6, followed by cultivation at 30 °C for 16 h. Bacterial pellets were resuspended in TES buffer (200 mM Tris-HCl pH 8.0, 0.65 mM EDTA, 0.5 M sucrose), after which periplasmic extracts were generated by osmotic shock in 0.25x TES at 4 °C overnight. Nanobodies were purified with Ni-NTA agarose beads (Qiagen), followed by desalting with PD MiniTrap G-25 columns (GE Healthcare Life Sciences) (ELISA experiments) or gel filtration with a HiLoad 16/600 Superdex 75 pg column (tissue culture experiments) in buffers containing 20 mM HEPES pH 7.4, 150 mM NaCl, and 10% glycerol.

To produce fluorescently labeled VHH$_{NP-1}$ or VHH$_{ASC}$ by sortase A labeling, 45 μM VHH-LPETG-His$_6$ was incubated with 475 μM GGGC-Alexa Fluor 647 and 20 μM His$_6$-tagged sortase A 7 m for 2 h[61]. Sortase A 7 m and unreacted VHHs were removed by depletion with Ni-NTA, followed by gel filtration on a Superdex 75 Increase 10/300 GL column. For experiments with primary cells, endotoxin was removed using the Pierce High Capacity Endotoxin Removal Spin Columns (Thermo Fischer Scientific).

**Antibodies.** The following antibodies were used (dilutions indicated): rabbit polyclonal anti-BID (Cell Signaling Technology Cat# 2002S, RRID:AB_10692485, 1:500), rabbit anti-caspase-3 clone D3R6Y (Cell Signaling Technology Cat# 14220, RRID:AB_2798429, 1:500), rabbit anti-cleaved caspase-3 (Asp175) clone 5A1E (Cell Signaling Technology Cat# 9664S, RRID:AB_2070042, 1:2000), rabbit anti-caspase-7 clone D2Q3L (Cell Signaling Technology Cat# 12827 T, RRID:AB_2687912, 1:500), mouse anti-caspase-8 clone 1C12 (Cell Signaling Technology Cat# 9746S, RRID:AB_2275120, 1:500), mouse anti-caspase-9 clone C9 (Cell Signaling Technology Cat# 9508S, RRID:AB_2068620, 1:500), rabbit anti-DFNA5/GSDME clone EPR19859 (Abcam Cat# ab215191, RRID:AB_2737000, 1:500), rabbit polyclonal anti-E-tag-HRP (Bethyl Cat# A190-133P, RRID:AB_345222, 1:10,000), mouse anti-GAPDH clone 0411 (Santa Cruz Biotechnology Cat# sc-47724, RRID:AB_627678, 1:1000), rabbit polyclonal anti-GSDMD (Atlas Antibodies Cat# HPA044487, RRID:AB_2678957, 1:500), rabbit anti-cleaved GSDMD$^{NT}$ clone EPR20829-40 (Abcam Cat# ab215203, RRID:AB_2916166, 1:500), mouse anti-HA.11 Epitope tag clone 16B12 (BioLegend Cat# 901503, RRID:AB_2565005, 1:50), mouse anti-HA-HRP clone 6E2 (Cell Signaling Technology Cat# 2999S, RRID:AB_1264166, 1:5,000), goat polyclonal anti-mouse IgG (H + L)-HRP (Invitrogen Cat# 31430, RRID:AB_228307, 1:5,000), goat polyclonal anti-rabbit IgG (H + L)-HRP (Invitrogen Cat# 31460, RRID:AB_228341, 1:5,000), highly cross-adsorbed goat polyclonal anti-mouse IgG (H + L)-Alexa Fluor™ 488 (Thermo Fisher Scientific Cat# A-11029, RRID:AB_2534088, 1:500), highly cross-adsorbed goat polyclonal anti-rabbit IgG (H + L)-Alexa Fluor™ Plus 647 (Thermo Fisher Scientific Cat# A32733, RRID:AB_2633282, 1:500), rabbit anti-PARP clone 46D11 (Cell Signaling Technology Cat# 9532S, RRID:AB_659884, 1:500), mouse anti-TOM20 clone 29 (BD Biosciences Cat# 612278, RRID:AB_399595, 1:500), and mouse anti-vinculin clone hVIN-1 (Sigma-Aldrich Cat# V9131, RRID:AB_477629, 1:1000).

**Small compound inhibitors and reagents.** The following small compound inhibitors and reagents were used: caspase-3/7 inhibitor I (Sigma), CRID3 (MCC-950) (Tocris), doxycycline (Biomol), LPS-EK Ultrapure (Invivogen), MG-132 (Selleckchem), Nigericin sodium salt (Biomol), Perfringolysin O (PFO) from *Clostridium perfringens* (CUSABIO), PMA (phorbol 12-myristate 13-acetate) (Sigma Aldrich), Roche cOmplete Mini protease Inhibitor Cocktail (Sigma Aldrich), staurosporine (Enzo), Vx-765/belnacasan (Selleckchem), Z-VAD(Ome)-FMK (MedChemExpress).

**Nanobody library generation.** To raise variable domains of heavy chain-only antibodies (VHHs) against human GSDMD, a male alpaca was immunized four times with 200 μg GSDMD using Imject Alum Adjuvant (Thermo Fisher Scientific) according to locally authorized protocols. The VHH plasmid library in the M13 phagemid vector pD (pJSC) was generated as described before[27,61]. In brief, RNA from peripheral blood lymphocytes was extracted and used as a template to generate cDNA using three sets of primers (random hexamers, oligo(dT), and primers specific for the constant region of the alpaca heavy chain gene). VHH coding sequences were amplified by PCR using VHH-specific primers, cut with AscI and NotI, and ligated into an M13 phagemid vector (pJSC) linearized with the same restriction enzymes. *E. coli* TG1 cells (Agilent) were electroporated with the ligation reactions and the obtained ampicillin-resistant colonies were harvested, pooled, and stored as glycerol stocks.

**Nanobody identification by phage display.** GSDMD-specific VHHs were obtained by phage display and panning with a protocol modified from Schmidt et al.[27]. *E. coli* TG1 cells containing the VHH library were infected with helper phage VCSM13 to produce phages displaying the encoded VHHs as pIII fusion proteins. Phages in the supernatant were purified and concentrated by precipitation. Phages presenting GSDMD-specific VHHs were enriched using chemically biotinylated GSDMD immobilized on Dynabeads MyOne Streptavidin T1 (Life Technologies). The retained phages were used to infect *E. coli* ER2738 and subjected to a second round of panning. 96 *E. coli* ER2837 colonies yielded in the second panning were grown in 96-well plates and VHH expression was induced with IPTG. VHHs leaked into the supernatant were tested for specificity using ELISA plates coated with control protein SUMO or SUMO-GSDMD. Bound VHHs were detected with HRP-coupled rabbit anti-E-Tag antibodies (1:10,000), and the chromogenic substrate tetramethylbenzidine (TMB) (Life Technologies). Reactions were stopped with 1 M HCl and absorption at 450 nm was recorded using a SpectraMax i3 instrument and the SoftMax Pro 6.3 Software (Molecular Devices). Positive candidates were sequenced and representative nanobodies were cloned into bacterial and mammalian expression vectors for further analysis.

**Nanobody ELISA.** To test nanobody candidates, SUMO-GSDMD or SUMO in PBS were immobilized on ELISA plates at a concentration of 1 μg/mL overnight. Subsequently, the immobilized antigens were incubated with the HA-tagged nanobodies in 10% FBS/PBS in a 10-fold dilution series ranging from 100 nM to 1 pM. The nanobodies were detected using the mouse anti-HA HRP antibody (1:5000) and developed using the chromogenic substrate TMB. The reaction was stopped using 0.5 M HCl, after which the absorption was measured at 450 nm using a SpectraMax i3 instrument and the SoftMax Pro 6.3 Software (Molecular Devices).

**LUMIER assay.** To test the functionality of VHHs in the reducing environment of the cellular cytosol, LUMIER assays were performed[62]. Renilla luciferase fusions of putative VHH targets were co-expressed with HA-tagged nanobodies in HEK293T cells. Nanobodies from cell lysates were immunoprecipitated with anti-HA antibodies and the co-purified luciferase activity determined as a readout for interaction under cytosolic conditions.

2.5 • 10$^5$ HEK293T cells per well were seeded into 24-well plates, and were co-transfected the next day with 0.25 μg pCAGGS VHH-HA expression vectors and 0.25 μg of pcDNA3.1-based expression vectors for the Renilla-fused bait proteins human GSDMD, GSDMD 4 A, GSDMD$^{NT}$ 4 A, GSDMD$^{CT}$, murine GSDMD, or the control human NLRP1$^{CARD}$ using PEI Max (Polysciences). High-binding Lumitrac 600 white 96-well plates (Greiner) were coated with 20 μg/mL of the mouse anti-HA.11 Epitope tag clone 16B12 antibody in PBS. One day post transfection, HEK293T cells were lysed in LUMIER lysis buffer (50 mM Hepes-KOH pH 7.9, 150 mM NaCl, 2 mM EDTA, 0.5% Triton X-100, 5% glycerol and Roche cOmplete Mini protease Inhibitor Cocktail) and bound to anti-HA-coated Lumitrac 600 plates for one hour to immunoprecipitate (IP) VHH-HA. After repeated washing, Renilla luciferase substrate coelenterazine-h was added to the IP well or lysate controls. Luminescence was measured using a SpectraMax i3 instrument and the SoftMax Pro 6.3 Software (Molecular Devices). The values plotted are the IP luminescence values normalized by the values of the lysate.

**Inflammasome activation.** To induce the human NLRC4 inflammasome, we employed 1.0 μg/mL *Bacillus anthracis* protective antigen (PA) to deliver recombinantly purified *Shigella flexneri* needle protein MxiH fused to *B. anthracis* LFn (LFn-MxiH, 0.1 μg/mL) into the cytosol for 1 h[63]. MxiH binds to human NAIP, which in turn initiates the oligomerization of NLRC4. NLRP3 is an indirect sensor for potassium efflux and perturbations of intracellular homeostasis[7]. To stimulate NLRP3, we primed the cells using 200 ng/mL ultrapure LPS for 3 h and activated NLRP3 by adding 10 μM nigericin (Nig), a potassium ionophore derived from *Streptomyces* hygroscopicus, for 1 h. Where indicated, caspase-1 activity was inhibited with 40 μM VX-765, or NLRP3 was inhibited with 2.5 μM CRID3 for 30 min before and during stimulation.

**Cell death quantification by LDH release (rupture).** To quantify ruptured pyroptotic cells, we measured the release of lactate dehydrogenase (LDH) activity into the supernatant using a chromogenic substrate. Of note, the tetrameric molecule is expected to be too big to leak through GSDMD pores. THP-1 cells were differentiated with 50 μg/mL PMA for 18 h, followed by a 24 h resting period; primary human macrophages were differentiated with M-CSF as described above. 3 • 10$^5$ cells were seeded in 24-well plates and intracellular VHH expression was induced with 1 μg/mL dox where inducible promoters were used. The NLRP3 and/or NLRC4 inflammasome was activated in OptiMEM as described above. The extracellular administration of recombinant VHHs in increasing concentrations (2, 20, 50, 100, and 200 μg/mL) occurred simultaneously with the inflammasome stimulus. To measure pyroptotic cell death in HEK293T cells triggered by gasdermin N-terminal domains, 5 • 10$^5$ cells per well were seeded into 24-well plates and co-transfected the next day with expression vectors for VHH-HA (0.5 μg) and GSDMD$^{NT}$, GSDME$^{NT}$ (0.25 μg), or empty vector using Lipofectamine 2000 (L2000) (Thermo Fisher Scientific). Supernatants were collected 24 h after the transient transfection.

LDH in the supernatants from either cell type was quantified using the LDH Cytotoxicity Detection kit (TaKaRa #MK401 or Roche #11 644 793 001) according to the manufacturer's instructions. Absorption at 492 nm was measured using a SpectraMax i3 instrument and the SoftMax Pro 6.3 Software (Molecular Devices). Medium background signals were subtracted from all values. Control samples, in which cells were lysed in 1% Triton X-100, were subsequently used to normalize LDH release.

**Cytokine quantification by HTRF.** To quantify IL-1β secretion, supernatants obtained as described for the LDH release assays were subjected to human IL-1β Homogeneous Time Resolved Fluorescence (HTRF) assays (Cisbio #62IL1BPEH) according to the manufacturer's instructions. Samples were excited at 340 nm and emissions at 616 nm and 665 nm were measured using a SpectraMax i3 instrument. IL-1β

levels were calculated by the SoftMax Pro 6.3 Software (Molecular Devices) based on the standard curve.

**Cell death quantification by DNA dye uptake (membrane integrity).** To quantify permeability of cellular membranes over time, we used DNA dyes DRAQ7, propidium iodide (PI), or SYTOX Green, which only intercalate into nuclear DNA and exhibits strong fluorescence when the plasma membrane is compromised by pores or damage. DRAQ7 was used to quantify membrane integrity over time with the Incucyte Live-Cell Imaging system (Sartorius) as described in the following, while the other reagents were used in confocal microscopy experiments as described in the figure legends.

$4 \cdot 10^4$ THP-1 cells per well were seeded in 96-well plates in the presence of PMA and the NLRP3 or NLRC4 inflammasome was activated as described above. Medium with stimuli was complemented with the non-cell permeable DNA dye DRAQ7 (100 nM) (Biolegend). DRAQ7 uptake was analyzed using the Incucyte Live-Cell Imaging system (Sartorius). The cells were recorded every 5 minutes for a total of 5 h using the Incucyte SX5 instrument, taking 4 images per well. The number of DRAQ7-positive nuclei (cell death count) and the cell confluency were analyzed using the Incucyte 2021 C software. For every single image, the cell death count was corrected by subtraction of the value at the start of the experiment. The corrected cell death count was further normalized to the cell confluency and average values from all 4 images were calculated and plotted over time.

**Cell death quantification by CellTiter-Blue assays (reducing activity).** CellTiter-Blue (CTB) assays were conducted to determine the reducing capacity and thus viability of untreated or stimulated cells using the CellTiter-Blue Reagent (Promega) according to the manufacturer's instructions. Cells in 96-well plates were treated as for LDH assays. Supernatants were aspirated and replaced with $100\,\mu L$ of the CTB reagent, followed by incubation at 37 °C for 1 h. Samples were excited with light at a wavelength of 560 nm and fluorescence was measured at 585 nm using a SpectraMax i3 instrument.

**Quantification of expression levels, inflammasome assembly, and caspase-3 cleavage by flow cytometry.** To quantify cells by flow cytometry, cells were harvested by trypsinization, fixed in 4% formaldehyde, and analyzed using a BD FACSCanto flow cytometer or, where indicated, with a BD LSRFortessa SORP flow cytometer using the following gating strategy: A homogenous population was gated from the FSC-A vs. SSC-A plot, followed by selection of single cells using SSC-A vs. SSC-W and FSC-A vs FSC-W plots. If not mentioned otherwise, cells expressing fluorescent transgenes or stained with fluorescent antibodies were gated using the area of the respective fluorescent channel. Transduction of primary human M-CSF macrophages with lentiviruses encoding the C1C-EGFP inflammasome reporter and VHH-HA was assessed by quantifying the fraction of cells positive for C1C-EGFP by flow cytometry. To estimate the total number of intact cells per sample, cells were treated identically, resuspended in the same volume, and measured by flow cytometry for a fixed time period of 30 s. The reduction of cells per volume served as an indirect indication for pyroptotic cell death. Caspase-1$^{CARD}$-EGFP (C1C-EGFP) recapitulates the recruitment of unprocessed caspase-1 to nascent ASC specks by homotypic interactions between caspase-1$^{CARD}$ and ASC$^{CARD}$ and can therefore be used as a fluorescent reporter for ASC speck and thus inflammasome assembly[30]. To quantify C1C-EGFP specks as a proxy for inflammasome assembly, we exploited that the peculiar redistribution of EGFP fluorescence from cytosolic to speck-associated yields a separate population of cells exhibiting higher fluorescence intensity, EGFP(H), and narrower width of the fluorescent signal, EGFP(W)[64]. We first gated cells positive for C1C-EGFP [EGFP(A)], and then plotted height against width of the C1C-EGFP signal[30,64]. For these experiments, $1 \cdot 10^5$ transduced primary macrophages or PMA-differentiated THP-1 derivatives in 24-wells were stimulated as described above. For the quantification of NLRP3 and NLRC4 inflammasome assembly in the presence of cytosolic VHH-EGFP fusions, PMA-differentiated THP-1 macrophages expressing both VHH-EGFP and C1C-mCherry were stimulated as described above. To prevent the loss of responding cells by caspase-1-dependent pyroptosis, the cells were stimulated in presence of $40\,\mu M$ VX. The fraction of specking C1C-mCherry positive cells was measured with a BD LSRFortessa SORP flow cytometer. Experiments were also performed in absence of VX and revealed that pyroptotic cells are lost during sample processing, while untreated and apoptotic cells could be analyzed by flow cytometry.

To measure cleaved caspase-3 in PMA-differentiated THP-1 macrophages, we performed experiments as described for apoptotic cell death. $3 \times 10^5$ THP-1 macrophages in wells of 24-well plates were treated with 1.0 µg/mL recombinant *B. anthracis* PA and 0.1 µg/mL LFn-MxiH for 1 h in presence of $40\,\mu M$ VX where indicated. Staurosporine is a non-selective inhibitor of several kinases and was added for 20 h as a positive control for intrinsic apoptosis and caspase-3 activation[65,66]. After fixation, cells were stained with rabbit anti-cleaved caspase-3 primary antibody (1:2000) and goat anti-rabbit Alexa Fluor Plus 647-coupled secondary antibody (1:500). The fraction of cells positive for cleaved caspase-3 was measured with a BD LSRFortessa SORP flow cytometer. All flow cytometry data was analyzed using the FlowJo 10.7.1 software.

**Confocal microscopy.** For live cell confocal microscopy experiments, PMA-differentiated THP-1 cells or GM-CSF differentiated primary human macrophages were cultured in 15 µ-slide 8 well Ibidi chambers ($9 \times 10^4$ cells) or black, clear bottom, TC treated PhenoPlate 96-well microscopy plates (Perkin Elmer) ($2 - 5 \times 10^4$ cells). Where indicated, cells were stained with CellMask Orange Plasma membrane stain (1:10,000, Thermo Fisher Scientific) at 37 °C for 10 min, followed by three washes with Opti-MEM. The NLRC4 inflammasome was activated with PA and LFn-MxiH for 1 h in imaging medium (RPMI with 10% FBS, $50\,\mu M$ 2-mercaptoethanol, 30 mM HEPES, no phenol red) using concentrations as described above. To stain endogenous proteins in microscopy samples, cells were seeded and treated as above, fixed in 4% formaldehyde in PBS for 20 minutes; where indicated, cells were stained with 5 µg/mL WGA AF647 and fixed again. To stain intracellular proteins, cells were permeabilized with 0.5% Triton X-100, and stained with Hoechst 33342 (Thermo Fisher Scientific) as well as rabbit anti-cleaved GSDMD$^{NT}$ antibody (1:500) + goat anti-rabbit IgG AF488 (1:1000), or mouse anti-TOM20 antibody (1:500) + goat anti-mouse IgG AF647 (1:1000) in PBS + 10% goat serum as indicated. Most images were recorded with the HC PL APO CS2 63x/1.20NA water objective on a Leica SP8 Lightning confocal microscope. Images in Fig. 4G as well as Figure S4A, S10, A-C, and movies S1/S2 were recorded with the HC PL APO CS2 63x/1.20NA water objective on a Leica Stellaris 8 microscope.

HEK293T cells constitutively expressing VHH-EGFP fusions were seeded in Ibidi chambers ($9 \times 10^4$ cells per well) coated with poly-L-lysine (mol wt 70,000-150,000) (Sigma Aldrich). They were transiently transfected with expression vectors for fusions of GSDMD and GSDMD$^{NT}$ variants with mCherry. In initial experiments, we employed the attenuating GSDMD mutant I104N to facilitate the observation of membrane-associated GSDMD$^{NT}$ as described before[2,31,33]. As the mutant largely behaved like WT GSDMD in our assays, we use WT GSDMD in later experiments. Where indicated, cells were co-transfected with expression vectors for emiRFP670 with a C-terminal CAAX motif (emiRFP670-CAAX). Emi-RFP670-CAAX is prenylated and anchors the fluorescent protein to the plasma membrane allowing us to assess membrane localization. 5 h post transfection, images were recorded at least every 10 minutes using the HC PL APO CS2 $63 \times 1.20$ water objective on a Leica SP8 Lightning confocal microscope (37 °C, 5% CO$_2$). Alternatively, cells were fixed 12 h post transfection, stained for DNA, and images recorded with the same microscopy setup.

**Image analysis.** Images were processed using ImageJ 2.3.0 software. To quantify the influx of fluorescent nanobodies, we used the cell detection tool of Imaris (Bitplane) to detect cell areas using the CMO channel, mostly containing correctly segmented cells, but occasionally clusters of cells (detection of cells without nucleus or vesicle staining; Cell Type = Cell Membrane; Cell Smallest Diameter = 12 μm; Cell Membrane Detail = 1 μm; Cell Filter Type = Local Contrast; Intensiy Manual Threshold = 4; Quality Manual Threshold = 0.090; Filter objects between 120 and 10,000 Voxels). We extracted the area, as well as the fluorescence intensity in the $VHH_{NP-1}$-AF647 channel. Cell areas were scored as $VHH^+$ if the mean intensity in the $VHH_{NP-1}$-AF647 channel was >80. The fraction of $VHH^+$ cell areas of the entire cell area was calculated and plotted. SYTOX green-positive nuclei were detected with the spot detection tool of Imaris (Estimated Diameter = 8 μm, Quality >10). Cells with distinct morphological features or fluorescence distribution were manually counted with the help of the counter function of Imaris. To quantify plasma membrane (PM) distribution of fluorescent $GSDMD^{NT}$ fusions, fluorescence intensity profiles along a line cutting the cell were analyzed for each cell. If the fluorescence only co-localized with the plasma membrane marker, localization was categorized as 'PM', when fluorescence was only found in the cytosol and dropped at the plasma membrane, localization was counted as 'cytosol', and when fluorescence above background was found in the cytosol, but fluorescence still increased at the PM, localization was scored as 'cytosol + PM'. When distribution of GSDMD-mNG_ins was quantified, 'clear plasma membrane localization' indicates that upon analysis of intensity profiles, the cell A) exhibits a distinct plasma membrane signal (clear peak of fluorescence at the rim of the cell, i.e., fluorescence appears as a relatively thin line), which requires that the focal plane cuts through the body of the cell and is not at the bottom or top of the cell; and that B) $GSDMD^{NT}$-mNG showed an equally crisp peak of fluorescence co-localizing with the plasma membrane staining (as apparent in intensity plot; $GSDMD^{NT}$-mNG staining appears as a thin line). This distinction was mostly used to exclude false positives, i.e., cells whose top or bottom were cut by the focal plane (no clear ring-like signal of plasma membrane marker), or cells with a large nucleus and very little cytoplasm next to it in the focal plane. In the latter case, the staining of $GSDMD^{NT}$-mNG sometimes also appeared ring-like, although the fluorescence was observed in a broader rim with a less sharp increase and no colocalization with the plasma membrane. Of note, the seemingly low fraction of responding cells can be explained by the lower number of cells with A) membrane staining perpendicular to the focal plane, and B) sufficient GSDMD-mNG or endogenous GSDMD expression and cleavage. SYTOX green intensity in Incucyte experiments was analyzed with the Incucyte software to extract the integrated fluorescence per field of view. To quantify SYTOX green intensity per nucleus, nuclei were detected with CellProfiler using the 'IdentifyPrimaryObjects' function (default settings, minimal radius = 10 pixel units, maximal radius = 40 pixel units) and the mean fluorescence intensity extracted with the 'MeasureObjectIntensity' function. To quantify localization of $VHH_{ASC}$-AF647 to C1C-EGFP specks or nuclei from Z stacks (5 slices, 2 μm apart), we first detected both structures using the Imaris spot detection routine (Specks: C1C-EGFP channel, Estimated Diameter = 2 μm, quality >10; nuclei: Hoechst channel, Estimated Diameter = 8 μm, quality >1) and enumerated C1C-EGFP specks with a $VHH_{ASC}$-AF647 intensity in the center >65 as well as nuclei with mean $VHH_{ASC}$-AF647 intensity >100. The fraction of $VHH_{ASC}$-AF647-positive nuclei was corrected by the fraction of positive cells observed in cells stimulated in the presence of $VHH_{NP-1}$ and VX (to correct variable background of cells that were negative for C1C-EGFP, but PI-/VHH-positive before treatment).

**Immunoblot.** To detect the presence and/or cleavage of proteins of interest, PMA-differentiated THP-1 cells were treated with 1.0 μg/mL recombinant *B. anthracis* PA and 0.1 μg/mL LFn-MxiH to activate the NLRC4 inflammasome. $3-4 \cdot 10^5$ cells (in 24-well plates) or $1.25 \cdot 10^6$ cells (in 6-well plates) were lysed in 100 μL or 300 mL RIPA buffer (50 mM Tris pH;7.4, 150 mM NaCl, 1% IGEPAL CA-630, 0.25% Na-deoxycholate, 2 mM EDTA, 0.1% SDS, Roche cOmplete Mini protease Inhibitor Cocktail), respectively. Samples were freshly supplemented with 4x SDS-PAGE buffer (yielding 50 mM Tris pH 6.8, 0.01% bromophenol blue, 10% glycerol, 2% SDS final) with or without 100 mM DTT, sheared and heated to 95 °C for 5 minutes. Proteins were separated by SDS-PAGE with 10% or 12% gels. Separated proteins were transferred to PVDF membranes (0.45 mm, Merck) by semi-dry transfer. All immunoblots were blocked in blocking buffer (5% non-fat dry milk (NFDM) in TBS with 0.05% Tween-20) for ≥2 h and probed with the following primary antibody dilutions: anti-BID (1:500), anti-caspase-3 (1:500), anti-caspase-7 (1:500), anti-caspase-8 (1:500), anti-caspase-9 (1:500), anti-GAPDH (1:1000), anti-GSDMD (1:500), anti-GSDME (1:500), anti-PARP (1:500), anti-vinculin (1:1000). Membranes were incubated with primary antibodies in blocking buffer at 4 °C overnight, washed, and probed with HRP-coupled secondary antibodies in blocking buffer (1:3000) for 2 h. Chemiluminescent signal was induced by Western Lightning® Plus-ECL (Perkin Elmer), except for immunoblots of BID, caspase-3, caspase-7, caspase-8, caspase-9, and GSDMD, which required Western Lightning Ultra (Perkin Elmer). The signal was detected using a Fusion Advancer imaging system (Vilber) and images were taken using the EvolutionCapt SL6 software (Vilber).

**Caspase-Glo activity and cell titer blue assay.** To quantify the activity of caspase-1, caspase-3/7, and caspase-8, $5 \cdot 10^4$ PMA-differentiated THP-1 cells cultured in 96-well plates were stimulated for 1 h with PA and LFn-MxiH as before, but in the presence of increasing concentrations of PFO (5, 10, 20, 30, 60, 120, 240, and 480 ng/mL) in optiMEM. To avoid additional activation of NLRP3 by potassium efflux through PFO pores, NLRC4 was in this case activated in the presence of NLRP3 inhibitor CRID3. Next, the cells plus supernatants were combined with an equal volume of the respective caspase-Glo reagent for 1 h according to the manufacturer's instructions (Promega). The mixture was then transferred to a Lumitrac 600 plate and the luminescence, resulting from cleavage of the caspase-3-specific peptide DEVD which renders a substrate available to luciferase, was measured using a SpectraMax i3 instrument and the SoftMax Pro 6.3 Software (Molecular Devices). Caspase-1-Glo assays were performed in the presence of 60 μM MG-132. The luminescence value of the negative control (OptiMEM plus caspase-Glo reagent control) was subtracted from all measured values. CellTiter-Blue (CTB) assays were conducted as described above to determine the reducing capacity and, thus, viability of untreated or stimulated cells with CellTiter-Blue Reagent (Promega). The value for the control well without cells was subtracted and fluorescence was normalized to cells expressing the control $VHH_{NP-1}$. The normalized CTB values were used to correct for differences in cell numbers between the different cell lines in the caspase-Glo assays. For the caspase-1 Glo assay, values were further normalized to MxiH-treated $VHH_{NP-1}$ samples.

### Reporting summary
Further information on research design is available in the Nature Portfolio Reporting Summary linked to this article.

## Data availability
The amino acid sequences of the described nanobodies $VHH_{GSDMD-1}$-$VHH_{GSDMD-6}$ are deposited in the Nanosaurus nanobody database (https://nanobodies.vib.be/) under the accession numbers NA-HNRC, NA-JLLX, NA-DREJ, NA-9KYU, NA-OE5I, and NA-B1BW. The values from all bar graphs displayed in this manuscript as well as uncropped scans of all immunoblots are supplied in the Source Data file. Source data are provided with this paper.

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

## Acknowledgements

We would like to acknowledge the support of the Flow Cytometry Core Facility of the Medical Faculty, University of Bonn, for their support, services and devices funded by the Deutsche Forschungsgemeinschaft (DFG, German Research Foundation) for project number 16372401, 387335189, 387333827, and 216372545. We are grateful to Gabor Horvath and the Microscopy Core Facility of the Medical Faculty at the University of Bonn for providing help, services and instrumentation supported by the Deutsche Forschungsgemeinschaft (DFG, German Research Foundation) for project number 388159768, and the Bundesministerium für Bildung und Forschung (BMBF, Federal Ministry of Education and Research)–ACCENT: Förderung von Advanced Clinician Scientist im Bereich Immunopathogenese und Organdysfunktion, Gehirn und Neurodegeneration–Förderkennzeichen: 01EO2107. We would like to thank Christina Martone and Jessica Ingram for providing the infrastructure for nanobody generation at the Whitehead Institute. The presented work was supported by the following funding agencies: Deutsche Forschungsgemeinschaft (DFG, German Research Foundation) grant SFB1403-414786233 (LDJS and FIS); Deutsche Forschungsgemeinschaft (DFG, German Research Foundation) grant TRR237-369799452 (YMT and FIS); Deutsche Forschungsgemeinschaft (DFG, German Research Foundation) Emmy Noether Program 322568668 (FIS); Deutsche Forschungsgemeinschaft (DFG, German Research Foundation) Germany's Excellence Strategy–EXC2151–390873048 (FIS and MG); Deutsche Forschungsgemeinschaft (DFG, German Research Foundation) grant GRK2168-272482170 (AK and MG).

## Author contributions

L.D.J.S.: Conceptualization, Data curation, Methodology, Investigation, Resources, Writing–original draft / review and editing, Visualization, Supervision; Y.M.T.: Investigation; L.-M.J.: Investigation, Visualization; S.D.: Investigation; S.C.B.: Investigation; S.N.: Methodology, Investigation; J.M.: Investigation; S.P.: Resources; E.H.: Investigation; A.K.: Conceptualization, Resources; A.A.: Resources; M.G.: Funding acquisition, Resources; H.L.P.: Funding acquisition, Resources; F.I.S.: Conceptualization, Data curation, Funding acquisition, Investigation, Methodology, Supervision, Visualization, Writing–original draft / review and editing.

## Funding

## Competing interests

FIS is cofounder and shareholder of Odyssey Therapeutics. LDJS, SN, AK, MG, HLP, and FIS are listed as inventors of a pending patent application on GSDMD nanobodies. The remaining authors declare no competing interests.
