## [Peer Review File · Nature Communications]

Antagonistic nanobodies implicate mechanism of GSDMD pore formation and unexpected therapeutic potentialEditorial note: This manuscript has been previously reviewed at another journal that is not operating a transparent peer review scheme. This document only contains reviewer comments and rebuttal letters for versions considered at *Nature Communications*.

REVIEWER COMMENTS

Reviewer #1 (Remarks to the Author):

The authors present an concise and well-written manuscript describing the inhibition of GSDMD pore formation using anti-GSDMD nanobodies. They have mostly addressed previous concerns by 2 reviewers, although the in vivo relevance of the findings remains to be shown.

Overall, the manuscript appears to be a good fit with Nature Communications, and complements the findings by Kopp et al. that were already published in the same journal.

I have thus only minor suggestions for improvement:

Figure 2: How long does the protection induced by VHH-GSDMD1/2 last? Can the authors add a timecourse of LDH release or PI uptake to Figure 2? And how does it compare to GSDMD deficiency in such a timecourse? Is it as good?

Figure 2M/N: The quality of the blots is poor, and loading is unequal. Please provide better blots, incl. loading controls for 2N.

Figure 2M/N: Why are GSDMD p20 bands seen in some lanes and not in others. In particular with VHH-GSDMD1, it is well visible. Does this VHH induce apoptosis?

Membrane insertion of VHH bound GSDMD-NT: Since C191 is farnesylated and thus quite important for membrane association and not only insertion, the fact that the C191 mutant is cytosolic cannot lead to the conclusion that VHH-bound GSDMD-NT is membrane inserted. It might just be associated. The authors could isolate membranes and wash them with high-salt buffer to remove associated proteins, or perform a protease protection experiment to

show that VHH-bound GSDMD still inserts.

Fig. 6C/D: Is the observed reduction in Casp1 activity caused by PFO treatment a consequence of PFO-induced death?